# Pseudo-Physics-Informed Neural Operators: Enhancing Operator Learning from Limited Data

**Keyan Chen\***                                                                          *u1466725@utah.edu*
*Kahlert School of Computing*
*University of Utah*

**Yile Li\***                                                                                  *yile.li@utah.edu*
*Kahlert School of Computing*
*University of Utah*

**Da Long**                                                                                  *da.long@utah.edu*
*Kahlert School of Computing*
*University of Utah*

**Zhitong Xu**                                                                            *u1502956@utah.edu*
*Kahlert School of Computing*
*University of Utah*

**Wei W. Xing**                                                                      *w.xing@sheffield.ac.uk*
*School of Mathematical and Physical Sciences*
*University of Sheffield*

**Jacob Hochhalter**                                                        *jacob.hochhalter@utah.edu*
*Department of Mechanical Engineering*
*University of Utah*

**Shandian Zhe**                                                                          *zhe@cs.utah.edu*
*Kahlert School of Computing*
*University of Utah*

**Reviewed on OpenReview:** *https://openreview.net/forum?id=5N1V25Rf7D*

## Abstract

Neural operators have shown great potential in surrogate modeling. However, training a well-performing neural operator typically requires a substantial amount of data, which can pose a major challenge in complex applications. In such scenarios, detailed physical knowledge can be unavailable or difficult to obtain, and collecting extensive data is often prohibitively expensive. To mitigate this challenge, we propose the Pseudo Physics-Informed Neural Operator (PPI-NO) framework. PPI-NO constructs a surrogate physics system for the target system using partial differential equations (PDEs) derived from simple, rudimentary physics principles, such as basic differential operators. This surrogate system is coupled with a neural operator model, using an alternating update and learning process to iteratively enhance the model's predictive power. While the physics derived via PPI-NO may not mirror the ground-truth underlying physical laws — hence the term "pseudo physics" — this approach significantly improves the accuracy of standard operator learning models in data-scarce scenarios, which is evidenced by extensive evaluations across five benchmark tasks and a fatigue modeling application. Our implementation is released at `https://github.com/BayesianAIGroup/PPI_NO`

# 1 Introduction

Operator learning, an important area for data-driven surrogate modeling, has made significant strides with the emergence of neural operators, which leverage the expressive power of neural networks. Notable examples include Fourier Neural Operators (FNO) (Li et al., 2020b), Deep Operator Net (DONet) (Lu et al., 2021) and other frameworks such as (Cao, 2021; Hao et al., 2023). FNO employs Fourier transform for global convolution and function transformation, while DONet introduces two sub-networks, the branch net and trunk net, to extract representations from the functional space and query locations, respectively, enabling predictions akin to attention mechanisms (Vaswani et al., 2017).

For trading for model capacity and performance, neural operators typically require a substantial amount of training data to perform optimally. This demand poses challenges, particularly in complex problems, where training data is limited and costly to acquire. In response, the field of physics-informed machine learning, including physics-informed neural networks (PINN) (Raissi et al., 2019), has shown promise by incorporating physical laws as soft constraints during training. This approach serves as a regularization technique, embedding a fundamental understanding of physics into the model to lessen its reliance on extensive data. Building on this idea, the concept of physics-informed neural operators (PINO) has emerged, which integrates physical laws as soft constraints to enhance operator learning while reducing data quantity. It has been used in (Wang et al., 2021; Li et al., 2021) for FNO and DONet training.

Despite the success of PINO, the necessity for a thorough understanding of the underlying physics can pose a significant hurdle, especially in complex applications such as in fracture mechanics and climate modeling. In those scenarios, the detailed physical knowledge is often unavailable or difficult to identify, and it is often prohibitively expensive to collect extensive data. To navigate these challenges while retaining the benefits of physics-informed learning, we propose the Pseudo Physics-Informed Neural Operator (PPI-NO). This framework bypasses the need for exhaustive physical comprehension by constructing a neural-network-based partial differential equation (PDE) that characterizes the target system directly from data. The neural PDE is then coupled with the neural operator for alternating updates and training, enabling iterative extraction, refinement and integration of physics knowledge to enhance operator learning. The contribution mainly lies in the following aspects:

- To our knowledge, PPI-NO is the first work to enhance a standard operator learning pipeline using physics directly learned from *limited data*, delivering superior accuracy without the need for in-depth physical understanding or extensive data collection.

- PPI-NO opens up a new paradigm of physics-informed machine learning where only rudimentary physics assumptions (in this case, the basic differential operations) are required rather than in-depth or rigorous expert knowledge, extending the spectrum of the physics-informed learning for experts of different levels.

- The effectiveness of PPI-NO is validated through extensive evaluations on five commonly used benchmark operator learning tasks in literature (Li et al., 2020b; Lu et al., 2022), including Darcy flow, nonlinear diffusion, Eikonal, Poisson and advection equations, as well as one application in fatigue modeling in fracture mechanics, where the ground-truth holistic PDE system is unknown.

# 2 Background

**Problem Formulation.** Operator learning seeks to approximate an operator that maps input parameters and/or functions to corresponding output functions. In many cases, operator learning rises in the context of solving partial differential equations (PDEs), where the operator corresponds to the solution operator of the PDE. Consider a PDE system:

$$\mathcal{N}[u](\mathbf{x}) = f(\mathbf{x}), \quad \mathbf{x} \in \Omega \times [0, \infty), \tag{1}$$

where $\mathbf{x}$ is a compact notation for the spatial and temporal coordinates, $\Omega$ is the spatial domain, $[0, \infty)$ is the temporal domain, $\mathcal{N}$ is a nonlinear differential operator, $u(\mathbf{x})$ is the solution function, and $f(\mathbf{x})$ is the source term. We aim to learn the solution operator of the PDE system, $\psi : \mathbb{F} \to \mathbb{U}$ where $\mathbb{F}$ and $\mathbb{U}$ are

two functional spaces, using a training dataset $\mathcal{D} = \{(\mathbf{f}_n, \mathbf{u}_n)\}_{n=1}^{N}$, which includes different instances of $u(\cdot)$ and $f(\cdot)$ sampled/discretized at a set of locations. Once the model is trained, it can be used to directly predict the solution function $u$ for new instances of the input $f$, offering a much more efficient alternative to running numerical solvers from scratch. However, the training dataset still needs to be generated offline using numerical solvers.

**Fourier Neural Operator (FNO)** (Li et al., 2020b) is a popular neural network architecture for operator learning, especially in solving PDEs. For a given discretized input function $\mathbf{f}$, FNO first employs a linear layer on each component of $\mathbf{f}$ at its respective sampling location, thereby lifting the input into a higher-dimensional channel space. The core of FNO is the Fourier layer, which performs a linear transformation followed by a nonlinear activation within the functional space, $h(\mathbf{x}) \leftarrow \sigma\left(\mathcal{W}h(\mathbf{x}) + \int \kappa(\mathbf{x} - \mathbf{x}')h(\mathbf{x}')d\mathbf{x}'\right)$, where $h(\mathbf{x})$ is the input to the Fourier layer, $\kappa(\cdot)$ the integration kernel, and $\sigma(\cdot)$ the activation function. The functional convolution is computed using the convolution theorem: $\int \kappa(\mathbf{x} - \mathbf{x}')h(\mathbf{x}')d\mathbf{x}' = \mathcal{F}^{-1}[\mathcal{F}[\kappa] \cdot \mathcal{F}[h]](\mathbf{x})$, where $\mathcal{F}$ and $\mathcal{F}^{-1}$ denote the Fourier and inverse Fourier transforms, respectively. FNO performs Fast Fourier Transform (FFT) on $h$, multiplies it with the discretized kernel in the frequency domain, and then applies the inverse FFT. After multiple Fourier layers, FNO employs another linear layer to project the latent channels to the original space for prediction.

**Deep Operator Network (DONet)** (Lu et al., 2021) is another prominent work in operator learning. The architecture of a DONet is structured into two components: a branch net and a trunk net, learning representations for the input functions and querying locations, respectively. Consider an input function $f(\mathbf{x}) \in \mathbb{F}$ sampled at $m$ locations $\{\mathbf{x}_1, \mathbf{x}_2, \cdots, \mathbf{x}_m\}$ and an output function $u \in \mathbb{U}$. The branch net receives the values $[f(\mathbf{x}_1), f(\mathbf{x}_2), \cdots, f(\mathbf{x}_m)]$ and outputs a feature representation $[b_1, b_2, \cdots, b_p]^\top \in \mathbb{R}^p$. Concurrently, the trunk network processes a querying location $\mathbf{x}$ and outputs another feature vector $[t_1, t_2, \cdots, t_p]^\top \in \mathbb{R}^p$. The output function value at $\mathbf{x}$ is predicted as a sum of products of the corresponding elements from the branch and trunk nets, $\mathcal{G}[f](\mathbf{x}) \approx \sum_{k=1}^{p} b_k t_k$, where $\mathcal{G}$ is the learned operator mapping $f$ to $u$.

**Physics-Informed Neural Operator (PINO)** (Wang et al., 2021; Li et al., 2021) has recently emerged as a promising approach to address the data scarcity issue in operator learning. PINO embeds physical laws — typically governing equations — into the learning process. The incorporation of physics not only enhances the model's adherence to ground-truth phenomena but also reduces its dependency on extensive training data. Mathematically, the integration of physics into the learning process can be viewed as adding a regularization term in the loss function. Let $\mathcal{L}_{\text{data}}$ represent the standard data-fitting loss term (*e.g.,* the mean squared error between the predicted and actual outputs), and the physics-informed term $\mathcal{L}_{\text{physics}}$ be the residual of the governing PDEs evaluated at the neural network's outputs. The loss function $\mathcal{L}$ for a PINO model is expressed as

$$\mathcal{L} = \mathcal{L}_{\text{data}} + \lambda \mathcal{L}_{\text{physics}},$$

where $\lambda$ is a weighting factor that balances the importance of data-fitting versus physics compliance. This approach encourages the model to learn solutions not only consistent with the provided data but also physically plausible.

## 3 Methodology

In the absence of physics knowledge (*i.e.,* PDE system equation 1), it is impossible to construct the physics loss term as in PINO. To address this challenge, we propose a "pseudo" physics-informed operator learning framework that extracts useful physics representation from data so as to enhance operator learning. This framework is motivated by relatively complex applications, where data is often costly and/or limited while the underlying physics is hard to fully understand.

### 3.1 Pseudo Physics System Learning

As the first step, we propose a novel approach to learn the physics system using scarce training data. Our key observation is that, *although the mapping from f to u can be intricate and may necessitate global information across the entire domain (e.g., in linear PDEs, u is an integration of the Green's function multiplied with f over the domain), the underlying PDE system equation 1 simplifies to a local combination of u and its*

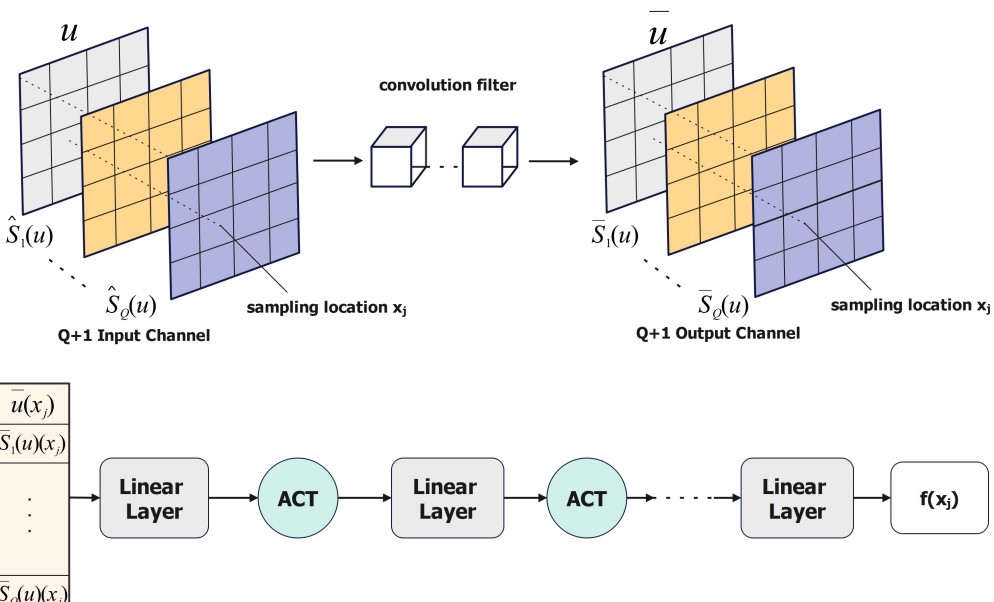

Figure 1: The illustration of "pseudo" physics representation network $\phi$. The input consists of $u$ and its finite difference derivative approximations $(\{\hat{S}_1(u), \ldots, \hat{S}_Q(u)\})$ across different sampling locations. The top row shows a convolution layer that aggregates local neighborhood to compensate the information loss caused by finite difference. The bottom row shows that $\phi$ uses fully connected layers at each sampling location to combine $u$ and its derivatives locally to predict $f$ at the same location.

*derivatives.* We therefore design a neural network $\phi$ to approximate the general form of $\mathcal{N}$,

$$\mathcal{N}[u](\mathbf{x}) \approx \phi\left(\mathbf{x}, u(\mathbf{x}), S_1(u)(\mathbf{x}), \ldots, S_Q(u)(\mathbf{x})\right), \tag{2}$$

where $\{S_j\}_{j=1}^Q$ are $Q$ derivative operators that we believe should be present in the system, such as $\partial_t u$, $\partial_{tt} u$, $\partial_{x_1} u$, $\partial_{x_2} u$, $\partial_{x_1 x_1} u$, $\partial_{x_1 x_2} u$, $\partial_{x_2 x_2} u$, and more.

The inherent local combinatorial nature of the PDE representation decouples the values of $u$ and its derivatives across different sampling locations, thereby significantly reducing the learning complexity and the amount of training data required. For instance, consider sampling the input function $f$ and output function $u$ on a $128 \times 128$ grid. A single pair of discretized input and output functions, denoted as $(\mathbf{f}, \mathbf{u})$, is typically insufficient for training a neural operator, because learning the mapping $f \to u$ requires abundant global information. However, this pair can be decomposed into $128 \times 128 = 16,384$ training data points across various (spatial and temporal) locations to train $\phi$ as outlined in equation 3. We use each point of the grid to construct a training sample. This decomposition provides rich information about the *local* relationships between those derivatives. Therefore, even with a small number of $(\mathbf{f}, \mathbf{u})$ pairs, we hypothesize that the learning of the PDE system $\mathcal{N}$ through our formulation in equation 3 can still yield promising accuracy in predicting $f(\cdot)$ as in equation 1.

We use an $L_2$ loss to estimate the parameters of $\phi$, which is defined as

$$\mathcal{L}_\phi = \sum_{n=1}^N \sum_{j=1}^M \Big[ \phi(\mathbf{x}_j, u_n(\mathbf{x}_j), S_1(u_n)(\mathbf{x}_j), \ldots, S_Q(u_n)(\mathbf{x}_j))$$
$$- f_n(\mathbf{x}_j) \Big]^2, \tag{3}$$

where $f_n(\cdot)$ and $u_n(\cdot)$ are the input and output functions in $n$-th training example, and $\{\mathbf{x}_1, \ldots \mathbf{x}_M\}$ are the locations at which we sample $f_n$ and $u_n$.

We use finite difference to obtain the derivatives of each $u_n$, namely, $S_k(u_n)$ $(1 \le k \le Q)$, and then pass these inputs to the network $\phi$ to compute the prediction. Specifically, we employ centered finite differences to

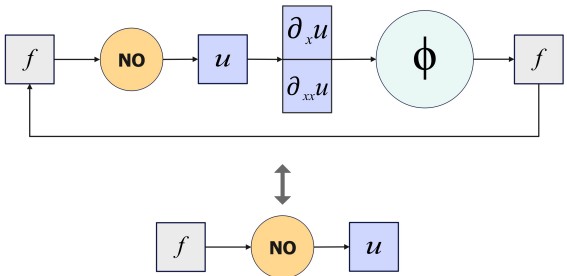

Figure 2: PPI-NO learning framework.

approximate the derivatives of $u$, a classical scheme with well-established accuracy and sample complexity. The approximation error scales as $O(h^2)$ for all derivative orders, where $h$ is the grid spacing. To achieve an error $\leq \epsilon$, the required number of grid points is $n = \Theta(\epsilon^{-d/2})$, where $d$ is the input dimension (Mitchell & Griffiths, 1980). It is worth-noting that while we can separate out training examples across grid points, these examples are not independent to each other. The effective sample size (ESS) for training our $\phi$-network is smaller than the total number of grid points. To compensate numerical errors of the finite difference and leverage inter-dependencies between the training examples, we incorporate a convolution layer to integrate the neighborhood information (see Fig. 1 top). Let $\hat{S}_k(u_n)$ represents the finite difference approximation of $S_k(u_n)$ ($1 \leq k \leq Q$); for $k = 0$, we define $\hat{S}_k(u_n) = u_n$. Each $\hat{S}_k(u_n)$ is treated as an input channel. After applying the convolution, the output at each sampling location $\mathbf{x}_j$ for channel $k$ is given by

$$\overline{S}_k(\mathbf{x}_j) = \sum_{c=0}^{Q} \sum_{\mathbf{x}_i \in \text{nei}(\mathbf{x}_j)} w_c(\Delta_{ij}) \hat{S}_c(\mathbf{x}_j), \tag{4}$$

where $\text{nei}(\cdot)$ denotes the neighborhood sampling locations defined in the convolution filter, $\Delta_{ij}$ is the relative distance between $\mathbf{x}_i$ and $\mathbf{x}_j$, and $w_c(\Delta_{ij})$ is the corresponding filter weight. Each $\overline{S}_k(\mathbf{x}_j)$ can be viewed as an interpolation, providing a new approximation of $S_k(\mathbf{x}_j)$ by incorporating all neighborhood values of $u_n(\mathbf{x}_j)$ and their finite difference derivative approximations. The convolution results are then passed into subsequent layers. The interpolation (filter) weights are jointly learned from data. This design allows us to integrate additional information into the inputs to facilitate the learning of $\phi$. Empirical results confirm that incorporating this convolution layer improves the prediction accuracy; refer to the ablation study in Section 5.2 and Table 7a. Next, at each sampling location $\mathbf{x}_j$, we employ fully connected layers (i.e., MLP) to combine all $\{\overline{S}_k(\mathbf{x}_j)\}_{k=0}^{Q}$ in an arbitrarily flexible way to predict $f(\mathbf{x}_j)$; see Fig. 1 bottom.

As a neural network, we note that the $\phi$-network, has sufficient representation power to approximate any PDE operator of the form

$$L(x, u(x), S_1(u)(x), \ldots, S_Q(u)(x)),$$

where $S_i$ denotes derivatives, e.g., $L = u_{xx} + uu_x + u^2 - 1 + \cos(u)$. Because most practical PDEs are smooth analytic functions of $u$ and its derivatives, the universal approximation theorem guarantees that such mappings can be accurately approximated by a feed-forward network of the following form:

$$f(x) = \sum_{i=1}^{N} a_i, \sigma(w_i^\top x + b_i),$$

given sufficient neurons and a non-polynomial activation (Cybenko, 1989) (Hornik, 1991). The sample complexity $m(\epsilon, \delta)$ is the smallest number of samples $m$ ensuring, with probability at least $1 - \delta$,

$$|R(h) - \hat{R}(h)| \leq \epsilon, \quad \forall h \in \mathcal{H},$$

where $R(h)$ and $\hat{R}(h)$ denote generalization and training errors, respectively. From computational learning theory (Shalev-Shwartz & Ben-David, 2014),

$$m(\epsilon, \delta) = O\left(\frac{d_{VC} + \log(1/\delta)}{\epsilon^2}\right),$$

where $d_{VC}$ is the VC dimension of the network. For a network with $W$ parameters and piecewise-linear activations (e.g., ReLU), $d_{VC} = O(W \log W)$; for smooth activations, $d_{VC} = O(W^2)$. Therefore, the learned neural network mapping $\phi : u \to f$, although black-box in nature, is expressive enough to encapsulate valuable physics knowledge inherent in the data employed for operator learning.

Our method can be readily adapted to other numerical approaches for derivative approximation. For instance, when functions are irregularly sampled, smooth function estimators such as kernel interpolation (Long et al., 2024), the RBF-FD method (Fornberg et al., 2013; Fornberg & Flyer, 2015), or Bayesian B-splines (Sun et al., 2022) can be used to estimate gradient information directly from data. These estimated gradients can then serve as inputs to our PDE neural network $\phi$ for further training.

### 3.2 Coupling Neural Operator with Pseudo Physics

Next, we leverage the pseudo physics laws embedded in the learned mapping $\phi : u \to f$ to enhance neural operator learning. Specifically, we use $\phi$ to reconstruct $f$ from the $u$ predicted by the neural operator. In this way, our approach uses the physics learned in the previous step to incorporate a reconstruction error into the learning of the neural operator parameters; see Fig. 2 for an illustration.

Initially, we train the neural operator $\psi : f \to u$ using the available training data, creating a preliminary model. This model is developed using FNO or DONet or other neural operators. The focus is to first establish a basic understanding of the relationship between $f$ and $u$ from the limited data. Next, the loss function for $\psi$ is augmented using the physics laws learned in the first step,

$$\mathcal{L} = \sum_{n=1}^{N} \mathcal{L}_2(\psi(f_n), u_n) + \lambda \cdot \mathbb{E}_{p(f')} \left[ \mathcal{L}_2(f', \phi(\psi(f'))) \right],$$

where the first term is the $\mathcal{L}_2$ loss for data fitting (as in the standard neural operator training), and the second term is the expected reconstruction error for the input function. The second term incorporates the physics laws embedded in $\phi(\cdot)$, and $\lambda$ is a weighting factor that balances the training data loss against the reconstruction error.

In practice, the expected reconstruction error does not have a closed form. One can sample a collection of $f'$ from the underlying distribution of the input function $p(\cdot)$, *e.g.,* a Gauss random field or Gaussian process, and then employ a Monte-Carlo approximation,

$$\mathcal{L} = \sum_{n=1}^{N} \mathcal{L}_2(\psi(f_n), u_n) + \lambda \frac{1}{N'} \sum_{n=1}^{N'} \mathcal{L}_2(f'_n, \phi(\psi(f'_n))), \tag{5}$$

where $N'$ is the number of input function samples. To be more specific, the PDE network $\phi$ acts purely as a model-based regularizer, without introducing any additional labeled data. In practice, it is realistic to sample as many as possible new input $f'$ from the same input distribution, but it is often not realistic to acquire all the corresponding ground-truth solutions $u'$. Thus, we sampled $f'$, pass it through the current operator network $\psi$ to obtain a predicted solution $\hat{u}'$, and then feed $\hat{u}'$ into the PDE network $\phi$ to produce a reconstructed source term $\tilde{f}'$. We use the discrepancy $\|\tilde{f}' - f'\|_2$ as a regularization term, as formalized in equation 5. It simply encourages $\psi$'s predictions to lie on the manifold of solutions that are consistent with the PDE representation $\psi$.

To enhance the operator learning process, the model is iteratively refined. In each iteration, we first fine-tune the neural operator $\psi$ with the pseudo physics $\phi$ fixed, and then fix $\psi$, fine-tune $\phi$ to refine the physics representation. This fine-tuning loop is carried out for multiple iterations, allowing for continuous improvement of the neural operator based on the refined physics representation. Our method is summarized in Algorithm 1.

---

**Algorithm 1** Pseudo-Physics-Informed NO

---

1: Train a preliminary NO $\psi$ with standard NO loss.
2: Train a preliminary psuedo physics network $\phi$ with loss $\mathcal{L}_\phi$ in equation 3.
3: **repeat**
4:    Sample $N'$ instances from the input function space.
5:    Fix pseudo physics network $\phi$, fine tune NO $\psi$ with the loss equation 5.
6:    Fix NO $\psi$, fine tune the pseudo physics network $\phi$ with the loss equation 5.
7: **until** Maximum iterations are done or convergence

---

## 4 Related Work

Operator learning is a rapidly evolving research field, with a variety of methods categorized as neural operators. Alongside FNO, several notable approaches have been introduced, such as low-rank neural operator (LNO) (Li et al., 2020b), multipole graph neural operator (MGNO) (Li et al., 2020a), multiwavelet-based NO (Gupta et al., 2021), and convolutional NOs (CNO) (Raonic et al., 2023). Deep Operator Net (DON) (Lu et al., 2021) is another popular approach, consisting of a branch network applied to input function values and a trunk network applied to sampling locations. The final prediction is obtained through the dot product of the outputs from the two networks. To enhance stability and efficiency, Lu et al. (2022) proposed replacing the trunk network with POD (PCA) bases. Recently, transformer architectures have also been employed to design neural operators, *e.g.,* (Cao, 2021; Li et al., 2022; Hao et al., 2023).

Physics-Informed Neural Networks (PINNs) (Raissi et al., 2019) mark a significant advancement in scientific machine learning. PINNs integrate physical laws directly into the learning process, making them effective for solving differential equations and understanding complex physical systems. This methodology is particularly beneficial in scenarios where data is limited or expensive to obtain. Li et al. (2021) introduced a dual-resolution approach that combines low-resolution empirical data with high-resolution PDE constraints. This method achieves precise emulation of solution operators across various PDE classes. In parallel, physics-informed DONet by Wang et al. (2021) incorporate regularization strategies enforcing physical law adherence into the training of DONets. Zanardi et al. (2023) presented an approach using PINO for simulations in non-equilibrium reacting flows. Lee et al. (2023) proposed opPINN, a framework combining physics-informed neural networks with operator learning for solving the Fokker-Planck-Landau (FPL) equation. Rosofsky et al. (2023) provided a review of applications of physics-informed neural operators. However, existing methods demand one must know the physics laws beforehand, which might not be feasible in many practical applications or complex systems. Our method offers a simple and effective framework, enabling the extraction of implicit physics laws directly from data, even when the data is scarce. Empirically, these pseudo physics laws have proven to be highly beneficial in enhancing the performance of operator learning, as demonstrated in Section 5. Many methods have been developed specifically for discovering differential equations from data, including SINDy (Brunton et al., 2016) and its variants (Schaeffer, 2017; Zhang & Ma, 2020; Lagergren et al., 2020), PINN-SR (Chen et al., 2021), Bayesian spline learning (Sun et al., 2022), and kernel-based equation discovery (Long et al., 2024). To ensure interpretability, these approaches typically assume a specific equation form and perform sparse selection from a set of candidate operators. In contrast to these methods, which prioritize interpretability, our approach focuses on enhancing the prediction accuracy of operator learning under limited data. To this end, we utilize a black-box neural network to represent PDEs. While this offers greater flexibility, it comes at the cost of reduced interpretability. Our method employs an alternating update strategy to jointly refine the PDE representation and improve operator learning.

Our work is also related to the cycle consistence framework (Zhu et al., 2017) for image-to-image translation. A critical difference is that cycle-consistence performs *unpaired* image-to-image translation, while our method aims for accurate paired translation (mapping). In cycle-consistence, the translation is viewed successfully as long as the translated images follow the target distribution. Hence, cycle-consistence has a much more relaxed objective. Another key difference is that our method aims to improve the learning of a function-to-function mapping with very limited data— that motivates us to learn a "pseudo" physics representation. The cycle-consistence framework relies on adversarial training which typically requires a large amount of data to obtain successful learning outcomes.

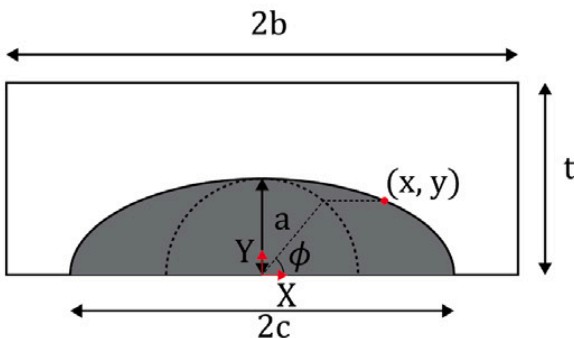

Figure 3: Example of semi-elliptic surface crack on a plates (Merrell et al., 2024).

## 5    Experiments

We tested on five commonly used benchmark operator learning problems in the literature (Li et al., 2020b; Lu et al., 2022), including *Darcy Flow*, *Nonlinear Diffusion*, *Eikonal*, *Poisson* and *Advection*. In addition, we examined our method in an application of fatigue modeling. The task is to predict the stress intensity factor (SIF) for semi-elliptical surface cracks on plates, given three geometric parameters that characterize the cracks (Merrell et al., 2024); see Fig. 3. The SIF plays a critical role in modeling crack growth by quantifying the stress state near the tip of a crack, and hence SIF computation and analysis are extremely important in fatigue modeling and fracture mechanics (Anderson & Anderson, 2005). The SIF computation is expensive, because it typically needs to run finite element method (FEM) or extended FEM with very fine meshes (Kuna, 2013). Due to the complex sequence of computational steps involved in SIF calculation, there is no holistic PDE that directly models the relationship between the geometric features and the SIF function. Instead, SIF computation typically relies on numerical methods and the extraction of local stress fields near the crack tip. The details about all the datasets are given in Section A of the Appendix.

We evaluated our method based on two widely used NO models, FNO and DONet. Note that our method is straightforward to implement on other NO models, such as attention based models.

For each operator learning benchmark, we simulated 100 examples for testing, and varied the number of training examples from $\{5, 10, 20, 30\}$, except for Advection, we ran with $\{20, 30, 50, 80\}$ training examples. For SIF prediction, which is much more challenging, we experimented with training size from $\{400, 500, 600\}$, and employed 200 test examples. We repeated the evaluation for five times, each time we randomly sampled a different training set. The input to the pseudo physics networks $\phi$ includes all the partial derivatives up to the 2nd order. For the pseudo physics neural network $\phi$ — see equation 3 — we tuned the kernel size from $\{(3, 3), (5, 5), (7, 7), (9, 9)\}$. The stride was set to 1 and padding was set to "same" to ensure the output shape does not change. In the subsequent fully connected layers, we chose the number of layers from $\{3, 4, 5, 6\}$, and the layer width from $\{16, 32, 64\}$. We used GeLU activation. For FNO, we set the number of modes to 12 and channels to 32 (in the lifted space). We varied the number of Fourier layers from $\{2, 3, 4\}$. For DONet, in all the cases except Darcy Flow, the trunk net and branch net were constructed as fully connected layers. We varied the number of layers from $\{2, 3, 4\}$ and the layer width was chosen from $\{30, 40, 50, 60\}$, with ReLU activation. For the case of Darcy flow, we found that DONet with only fully connected layers exhibited inferior performance. To address this, we introduced convolution layers into the branch net. We selected the number of convolution layers from $\{3,5,7\}$, and employed batch normalization and leaky ReLU after each convolution layer. To incorporate the learned pseudo physics representation into the training of FNO or DONet, we randomly sampled 200 input functions to construct the second loss term in equation 5. We set the maximum number of iterations to 10 and selected the weight $\lambda$ from $[10^{-1}, 10^2]$. All the models were implemented by PyTorch (Paszke et al., 2019), and optimized with ADAM (Kingma & Ba, 2014). The learning rate was selected from $\{10^{-4}, 5 \times 10^{-4}, 10^{-3}\}$. The number of epochs for training or fine-tuning FNO, DONet and pseudo physics network $\phi$ was set to 500 to ensure convergence.

Table 1: Relative $L_2$ error in five operator learning benchmarks, where "PPI" is short for "Pseudo-Physics Informed". The results were averaged from five runs.

(a) *Darcy flow*

| Training size | 5 | 10 | 20 | 30 |
|---|---|---|---|---|
| FNO | $0.4915 \pm 0.0210$ | $0.3870 \pm 0.0118$ | $0.2783 \pm 0.0212$ | $0.1645 \pm 0.0071$ |
| PPI-FNO | $0.1716 \pm 0.0048$ | $0.0956 \pm 0.0084$ | $0.0680 \pm 0.0031$ | $0.0642 \pm 0.0010$ |
| Error Reduction | 65.08% | 75.29% | 75.56% | 60.97% |
| DONet | $0.8678 \pm 0.0089$ | $0.6854 \pm 0.0363$ | $0.5841 \pm 0.0279$ | $0.5672 \pm 0.0172$ |
| PPI-DONet | $0.5214 \pm 0.0543$ | $0.3408 \pm 0.0209$ | $0.2775 \pm 0.0224$ | $0.2611 \pm 0.0084$ |
| Error Reduction | 39.91% | 50.27% | 52.49% | 53.96% |

(b) *Nonlinear diffusion*

| Training size | 5 | 10 | 20 | 30 |
|---|---|---|---|---|
| FNO | $0.2004 \pm 0.0083$ | $0.1242 \pm 0.0046$ | $0.0876 \pm 0.0061$ | $0.0551 \pm 0.0021$ |
| PPI-FNO | $0.0105 \pm 0.0016$ | $0.0066 \pm 0.00023$ | $0.0049 \pm 0.00037$ | $0.0038 \pm 0.00039$ |
| Error Reduction | 94.76% | 94.68% | 94.40% | 93.10% |
| DONet | $0.3010 \pm 0.0119$ | $0.2505 \pm 0.0057$ | $0.1726 \pm 0.0076$ | $0.1430 \pm 0.0036$ |
| PPI-DONet | $0.1478 \pm 0.0126$ | $0.1161 \pm 0.0124$ | $0.1032 \pm 0.0059$ | $0.0842 \pm 0.0041$ |
| Error Reduction | 50.89% | 53.65% | 40.20 % | 41.11% |

(c) *Eikonal*

| Training size | 5 | 10 | 20 | 30 |
|---|---|---|---|---|
| FNO | $0.2102 \pm 0.0133$ | $0.1562 \pm 0.0098$ | $0.0981 \pm 0.0022$ | $0.0843 \pm 0.0020$ |
| PPI-FNO | $0.0678 \pm 0.0026$ | $0.0582 \pm 0.0043$ | $0.0493 \pm 0.0023$ | $0.0459 \pm 0.0010$ |
| Error Reduction | 67.74% | 62.74% | 49.74% | 45.55% |
| DONet | $0.3374 \pm 0.0944$ | $0.1759 \pm 0.0065$ | $0.1191 \pm 0.0047$ | $0.1096 \pm 0.0037$ |
| PPI-DONet | $0.1302 \pm 0.0127$ | $0.0907 \pm 0.0093$ | $0.0714 \pm 0.0011$ | $0.0700 \pm 0.0007$ |
| Error Reduction | 61.41% | 48.43% | 40.05% | 36.13% |

(d) *Poisson*

| Training size | 5 | 10 | 20 | 30 |
|---|---|---|---|---|
| FNO | $0.2340 \pm 0.0083$ | $0.1390 \pm 0.0007$ | $0.0895 \pm 0.0008$ | $0.0698 \pm 0.0014$ |
| PPI-FNO | $0.1437 \pm 0.0062$ | $0.0771 \pm 0.0018$ | $0.0544 \pm 0.0009$ | $0.0458 \pm 0.0003$ |
| Error Reduction | 38.59% | 44.53% | 39.22% | 34.38% |
| DONet | $0.6142 \pm 0.0046$ | $0.5839 \pm 0.0090$ | $0.5320 \pm 0.0028$ | $0.5195 \pm 0.0040$ |
| PPI-DONet | $0.5275 \pm 0.0037$ | $0.5001 \pm 0.0042$ | $0.4450 \pm 0.0010$ | $0.4258 \pm 0.0040$ |
| Error Reduction | 14.12% | 14.35% | 16.35% | 18.04% |

(e) *Advection*

| Training size | 20 | 30 | 50 | 80 |
|---|---|---|---|---|
| FNO | $0.4872 \pm 0.0097$ | $0.4035 \pm 0.0086$ | $0.3019 \pm 0.0085$ | $0.2482 \pm 0.0059$ |
| PPI-FNO | $0.3693 \pm 0.0099$ | $0.3224 \pm 0.0123$ | $0.2236 \pm 0.0075$ | $0.1698 \pm 0.0075$ |
| Error Reduction | 24.20% | 20.10% | 25.94% | 31.59% |
| DONet | $0.5795 \pm 0.0045$ | $0.4810 \pm 0.0092$ | $0.3882 \pm 0.0086$ | $0.3164 \pm 0.0072$ |
| PPI-DONet | $0.3630 \pm 0.0112$ | $0.2897 \pm 0.0097$ | $0.2629 \pm 0.0053$ | $0.2120 \pm 0.0065$ |
| Error Reduction | 37.36% | 39.77% | 32.28% | 33.00% |

## 5.1 Predictive Performance

Table 2: SIF prediction error for plate surface cracks in fatigue modeling.

| Training size | 400 | 500 | 600 |
|---|---|---|---|
| FNO | $0.1776 \pm 0.0150$ | $0.1695 \pm 0.0090$ | $0.1122 \pm 0.0094$ |
| PPI-FNO | $0.1166 \pm 0.0064$ | $0.1151 \pm 0.0093$ | $0.0850 \pm 0.0060$ |
| Error Reduction | 34.35% | 32.09% | 24.24% |
| DONet | $0.5318 \pm 0.0095$ | $0.5155 \pm 0.0200$ | $0.4037 \pm 0.0331$ |
| PPI-DONet | $0.3490 \pm 0.0034$ | $0.3468 \pm 0.0074$ | $0.3299 \pm 0.0066$ |
| Error Reduction | 34.37% | 32.73% | 18.28% |

Table 3: Relative $L_2$ error of using the learned back-box PDE network equation 3 to predict the input function $f$.

(a) Training size=10

| Benchmark | MLP | Ours |
|---|---|---|
| Darcy Flow | $0.1819 \pm 0.0026$ | $\mathbf{0.1392 \pm 0.0080}$ |
| Nonlinear Diffusion | $0.0660 \pm 0.0069$ | $\mathbf{0.0233 \pm 0.0005}$ |
| Eikonal | $0.0144 \pm 0.0009$ | $\mathbf{0.0108 \pm 0.0006}$ |

(b) Training size=30

| Benchmark | MLP | Ours |
|---|---|---|
| Darcy Flow | $0.1413 \pm 0.0013$ | $\mathbf{0.0688 \pm 0.0032}$ |
| Nonlinear Diffusion | $0.0463 \pm 0.0022$ | $\mathbf{0.0163 \pm 0.0002}$ |
| Eikonal | $0.0070 \pm 0.00005$ | $\mathbf{0.0052 \pm 0.0002}$ |

We reported the average relative $L_2$ error and the standard deviation (with and without using pseudo physics informed learning) in Table 1 and Table 2. The model trained with the pseudo physics network (see equation 5) is denoted as PPI-FNO or PPI-DONet, short for Pseudo Physics Informed FNO/DONet. As shown, across all the cases, with our pseudo physics informed approach, the prediction error of both FNO and DONet undergoes a large reduction. For instance, across all training sizes in *Darcy Flow* and *nonlinear diffusion*, PPI-FNO reduces the relative $L_2$ error of the ordinary FNO by over 60% and 93%, respectively. In *Darcy Flow* with training sizes 10 to 30, PPI-DONet reduces the error of the ordinary DONet by over 50%. In SIF prediction, our method applied to both FNO and DONet reduced the error by over 30% for training sizes of 400 and 500. Even the minimum reduction across all cases achieves 14.12% (PPI-DONet over DONet on *Poisson* with training size 5).

Together these results demonstrate the strong positive impact of the learned physics by our neural network model $\phi$ specified in Section 3.1. Although it remains opaque and non-interpretable, it encapsulates valuable knowledge that substantially enhances the performance of operator learning with limited data.

Next, we assessed the accuracy of the learned physics laws by examining the relative $L_2$ error in predicting the source functions $f$ from $\phi$ (see equation 3). We tested on *Darcy Flow*, *nonlinear diffusion*, and *Eikonal*. We compared a baseline method that removes the convolution layer of $\phi$, leaving only the fully connected layers, namely MLP (see Fig 1 bottom). The results are reported in Table 3. It can be observed that in nearly every case, adding a convolution layer indeed significantly improves the accuracy of $\phi$. This improvement might be attributed to the convolution layer's ability to integrate neighboring information and compensate for the information loss introduced by finite difference in approximating the derivatives. We also experimented with multiple convolution layers, but the improvement was found to be marginal.

In addition, we found the operator learning improvement is relatively *robust* to the accuracy of our physics representation $\phi$. For instance, on *Darcy Flow* with training size 5 and 10, the relative $L_2$ error of $\phi$ network is 0.2285 and 0.1392, which is significantly bigger than with training size 30 where the relative $L_2$ error is 0.0688. Yet the error reduction upon FNO (see Table 1a) under all the three training sizes is above 60%. The error reduction upon DONet is 40% for training size 5 and over 50% for training size 10 and 30. The results imply that even roughly capturing the underlying physics (with $\phi$) can substantially boost the operator learning performance.

Table 4: Relative $L_2$ error in five operator learning benchmarks with richer data, where "PPI" is short for "Pseudo-Physics Informed". The results were averaged from five runs.

(a) *Darcy flow*

| Training size | 600 | 1000 |
|---|---|---|
| FNO | $0.0093 \pm 0.00015$ | $0.0079 \pm 0.00018$ |
| PPI-FNO | $0.0087 \pm 0.00040$ | $0.0082 \pm 0.00039$ |
| DONet | $0.0540 \pm 0.00064$ | $0.0446 \pm 0.00023$ |
| PPI-DONet | $0.0415 \pm 0.00077$ | $0.0362 \pm 0.00049$ |

(b) *Nonlinear diffusion*

| Training size | 600 | 1000 |
|---|---|---|
| FNO | $0.0035 \pm 0.0007$ | $0.0028 \pm 0.0004$ |
| PPI-FNO | $0.0047 \pm 0.00102$ | $0.0042 \pm 0.00096$ |
| DONet | $0.0222 \pm 0.00020$ | $0.0187 \pm 0.00023$ |
| PPI-DONet | $0.0379\pm 0.00088$ | $0.0368 \pm 0.00093$ |

(c) *Eikonal*

| Training size | 600 | 1000 |
|---|---|---|
| FNO | $0.0193 \pm 0.00017$ | $0.0148 \pm 0.00009$ |
| PPI-FNO | $0.0199 \pm 0.00016$ | $0.0160 \pm 0.00009$ |
| DONet | $0.0460 \pm 0.00021$ | $0.0411 \pm 0.00038$ |
| PPI-DONet | $0.0475\pm 0.00047$ | $0.0436 \pm 0.00034$ |

(d) *Poisson*

| Training size | 600 | 1000 |
|---|---|---|
| FNO | $0.0045 \pm 0.00006$ | $0.0037 \pm 0.00005$ |
| PPI-FNO | $0.0046 \pm 0.00005$ | $0.0040 \pm 0.00006$ |
| DONet | $0.1786 \pm 0.00398$ | $0.1719 \pm 0.00643$ |
| PPI-DONet | $0.1409 \pm 0.00292$ | $0.1373 \pm 0.00136$ |

(e) *Advection*

| Training size | 600 | 1000 |
|---|---|---|
| FNO | $0.0943 \pm 0.00177$ | $0.0768 \pm 0.00182$ |
| PPI-FNO | $0.0819 \pm 0.00092$ | $0.0695 \pm 0.00092$ |
| DONet | $0.0913 \pm 0.00074$ | $0.0748 \pm 0.00098$ |
| PPI-DONet | $0.0732 \pm 0.00143$ | $0.0626 \pm 0.00107$ |

Table 5: Relative $L_2$ error in Poisson and Advection operator learning benchmarks, where "PPI" is short for "Pseudo-Physics Informed" and "PI" is truly "Physics Informed". The results were averaged from five runs.

(a) *Poisson*

| Training size | 5 | 10 | 20 | 30 |
|---|---|---|---|---|
| FNO | $0.2340 \pm 0.0083$ | $0.1390 \pm 0.0007$ | $0.0895 \pm 0.0008$ | $0.0698 \pm 0.0014$ |
| PPI-FNO | $0.1437 \pm 0.0062$ | $0.0771 \pm 0.0018$ | $0.0544 \pm 0.0009$ | $0.0458 \pm 0.0003$ |
| PI-FNO | $\mathbf{0.1433 \pm 0.0104}$ | $\mathbf{0.0718 \pm 0.0015}$ | $\mathbf{0.0504 \pm 0.0009}$ | $\mathbf{0.0429 \pm 0.0004}$ |

(b) *Advection*

| Training size | 20 | 30 | 50 | 80 |
|---|---|---|---|---|
| FNO | $0.4872 \pm 0.0097$ | $0.4035 \pm 0.0086$ | $0.3019 \pm 0.0085$ | $0.2482 \pm 0.0059$ |
| PPI-FNO | $0.3693 \pm 0.0099$ | $0.3224 \pm 0.0123$ | $0.2236 \pm 0.0075$ | $0.1698 \pm 0.0075$ |
| PI-FNO | $\mathbf{0.3628 \pm 0.0082}$ | $\mathbf{0.3205 \pm 0.0121}$ | $\mathbf{0.2222 \pm 0.0074}$ | $\mathbf{0.1668 \pm 0.0057}$ |

For a further assessment, we conducted a fine-grained evaluation by visualizing the predictions and point-wise errors made by each method. In Fig. 4a and 4b, we showcased the predictions and point-wise errors using PPI-DONet for *Darcy Flow*, PPI-FNO for *nonlinear diffusion*, respectively. Additional examples of predictions and point-wise errors are provided in Fig. 5a, 5b, 6a, 6b, and 7.

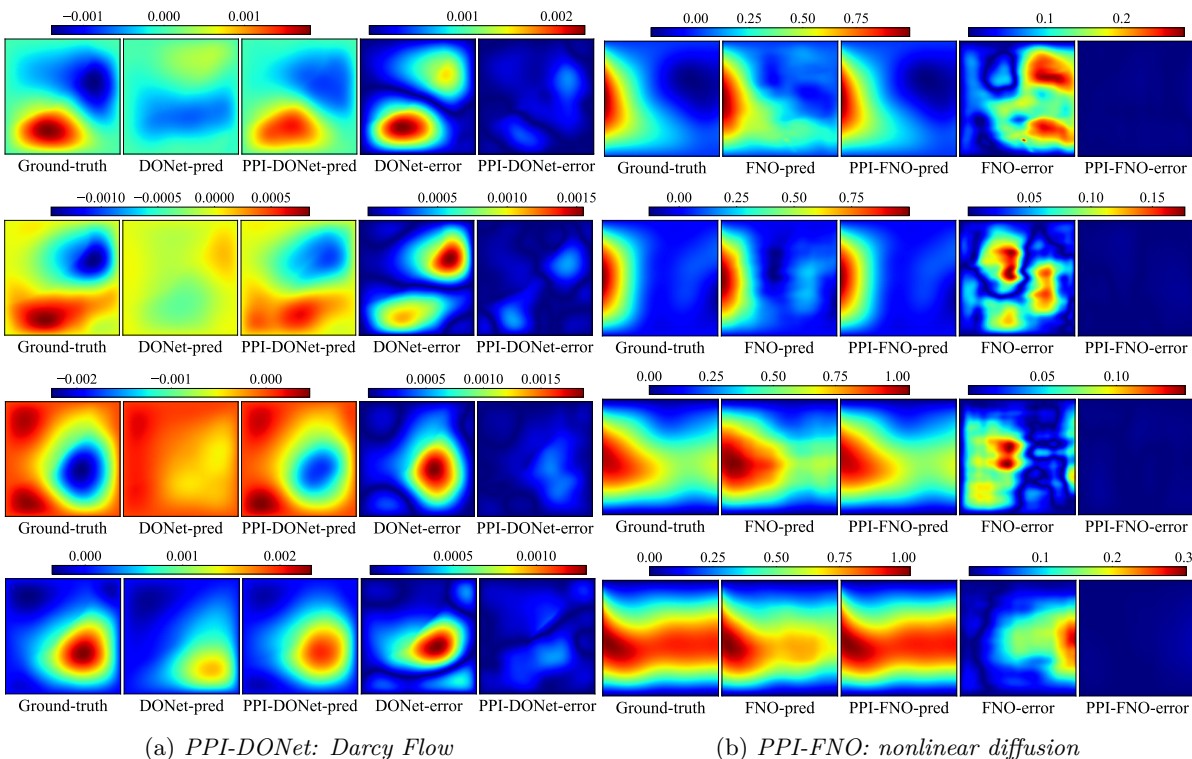

(a) *PPI-DONet: Darcy Flow*        (b) *PPI-FNO: nonlinear diffusion*

Figure 4: Examples of the prediction and point-wise error of PPI-DONet and PPI-FNO on *Darcy Flow* and *nonlinear diffusion*, respectively. From top to bottom, the models were trained with 5, 10, 20, 30 examples.

It is evident that without the assistance of the pseudo physics laws learned by our method, the ordinary DONet and FNO frequently missed crucial local structures, sometimes even learning entirely incorrect structures. For example, In Fig. 4a the first row, DONet missed one mode, while in the second and third row of Fig. 4a, DONet failed to capture all the local modes. After incorporating the learned physics, DONet (now denoted as PPI-DONet; see the third column) successfully captures all the local modes, including their shapes and positions. Although not all the details are exactly recovered, the point-wise error is substantially reduced, particularly in those high error regions of the ordinary DONet; see the fourth column of Fig. 4a. In another instance, as shown in Fig. 4b, where the ordinary FNO (second column) captured the global shape of the solution, but the mis-specification of many local details led to large point-wise errors across many regions (fourth column). In contrast, PPI-FNO (third column) not only identified the structures within the solution but also successfully recovered the details. As a result, the point-wise error (fifth column) was close to zero everywhere. Additional instances can be found in Fig. 5a, the first three rows illustrate that ordinary FNO (trained with 5, 10, and 20 examples, respectively) estimates an entirely incorrect structure of the solution, indicating that the training data is insufficient for FNO to capture even the basic structure of the solution. In contrast, after joint training with our physics network from the same sparse data, PPI-FNO accurately figured out the solution structures and yielded a substantial reduction in point-wise error across nearly everywhere. The point-wise error became uniformly close to zero. With 30 examples, the ordinary FNO was then able to capture the global structure of the solution, but the details in the bottom left, bottom right, and top right corners were incorrectly predicted. In comparison, PPI-FNO further recovered these details accurately.

Collectively, these results demonstrate that the pseudo physics extracted by our method not only substantially boosts the overall prediction accuracy but also better recovers the local structures and details of the solution.

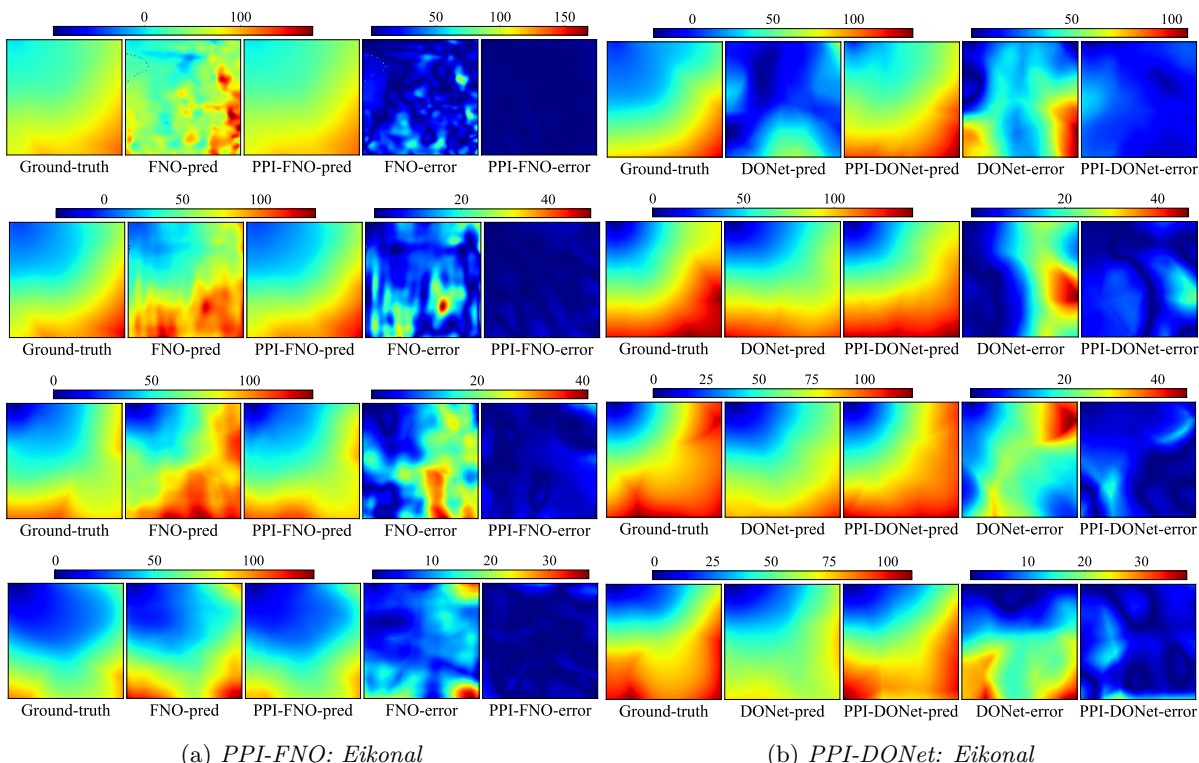

(a) *PPI-FNO: Eikonal*        (b) *PPI-DONet: Eikonal*

Figure 5: Examples of the prediction and point-wise error of PPI-FNO and PPI-DONet on *Eikonal*. From top to bottom, the models were trained with 5, 10, 20, 30 examples.

## 5.2 Ablation Studies

**Step-by-step approach: system-identification-then-neural-operator.** We additionally evaluate a stepwise pipeline in which a standard system-identification method is first used to infer an explicit PDE model, and the resulting equation is then used to regularize neural-operator training. However, system identification methods targeting interpretability typically require stronger assumptions and more prior knowledge than our black-box PDE learner. For instance, the widely used SINDy framework (Brunton et al., 2016) assumes a linear structure and depends on a predefined operator dictionary—whose choice is crucial and can easily cause model misspecification.

In contrast, our black-box PDE representation is more flexible and assumption-light: we only specify a set of basic derivatives, while their nonlinear combinations (e.g., $\sin(u)$, $uu_x$, $u_{xx}^2$, $u_x u_y$) are learned by a neural network. This design inevitably sacrifices interpretability in exchange for flexibility. Furthermore, our joint training of the neural operator and the PDE representation $\phi$ allows both components to mutually enhance each other's performance.

To evaluate the "system-identification–then–neural-operator" alternative, we ran SINDy to first identify an approximate governing equation, used it to generate paired synthetic data, and then trained the FNO on the combined real and synthetic samples. The operator dictionary matched the derivative set used in our method. For comparison, we also ran our black-box $\phi$-network to generate 200 synthetic examples. The resulting test errors of the neural operator under varying training data sizes are reported in Table 6.

We observe that applying SINDy before training FNO improves prediction accuracy only in the Darcy flow task with 5, 10 and 20 training examples; in all other cases, performance actually deteriorates compared to training FNO alone. This suggests that the recovered equations, while interpretable, may still be inaccurate enough that the synthetic samples they generate can adversely affect FNO training.

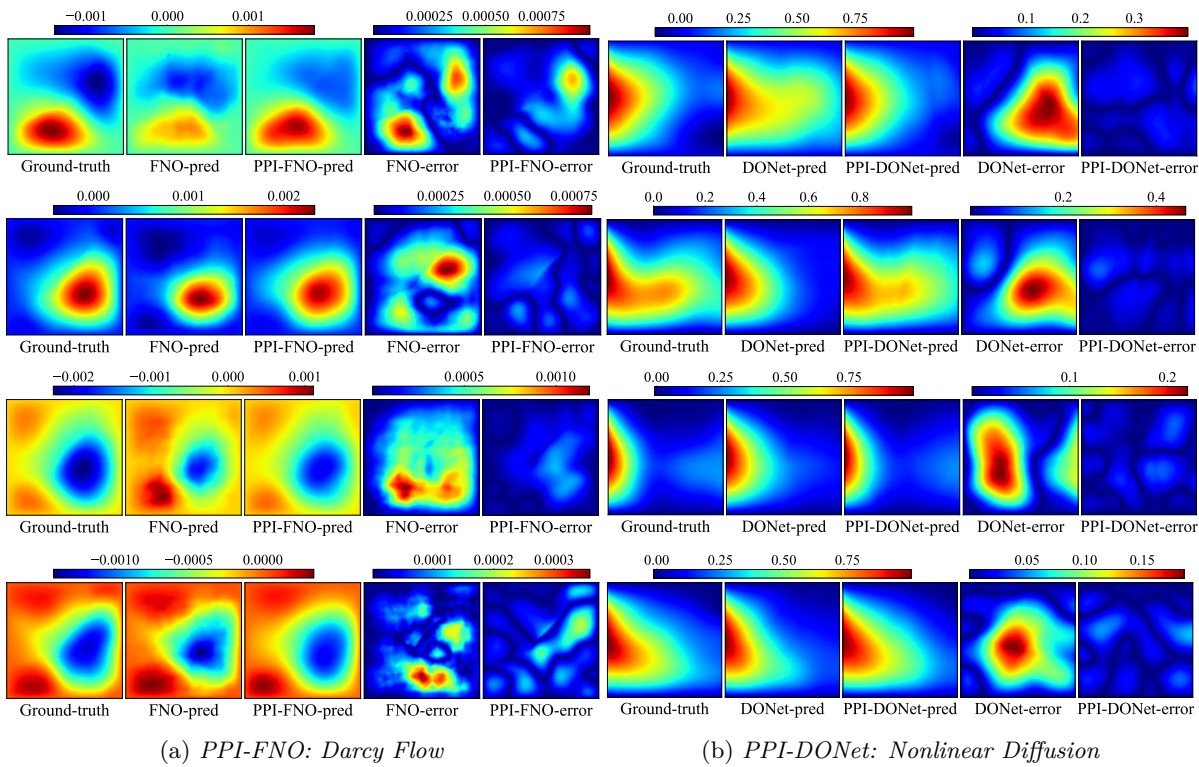

|  |  |
|---|---|
| (a) *PPI-FNO: Darcy Flow* | (b) *PPI-DONet: Nonlinear Diffusion* |

Figure 6: Examples of the prediction and point-wise error of PPI-FNO and PPI-DONet on *Darcy Flow* and *Nonlinear diffusion*, respectively. From top to bottom, the models were trained with 5, 10, 20, 30 examples.

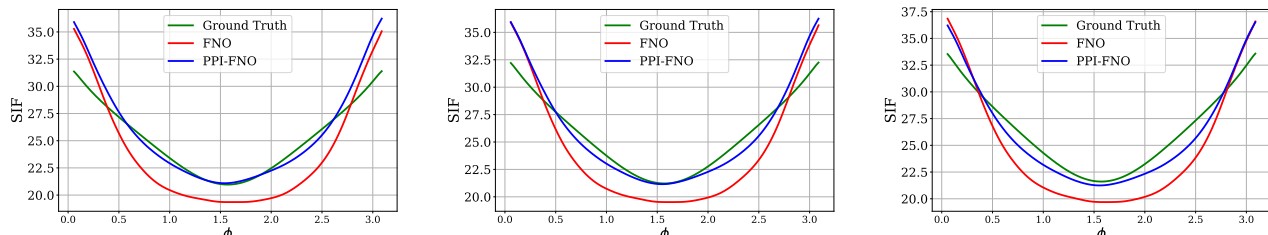

Figure 7: Examples of SIF prediction of FNO and PPI-FNO trained with 600 examples.

In contrast, the "$\phi$-network–then–FNO pipeline"—where our $\phi$-network first generates synthetic data, followed by FNO training—consistently improves results. Although the $\phi$-network is a black-box model, it can more accurately approximate the underlying PDE structure, leading to more useful synthetic samples. Nevertheless, this sequential approach still underperforms our "jointly trained $\phi$ + FNO" model, demonstrating that mutual optimization between the two components further enhances FNO's predictive accuracy.

**Training with rich data.** Although our method is designed to enhance operator learning with limited data, we also evaluated its performance when the training data is abundant. Specifically, we increased the training size of the five operator learning benchmarks to 600 and 1000 examples. The results, presented in Table 4, show that the accuracy of the standard NO and our method, PPI-NO, becomes comparable. For *Darcy flow* and *Advection*, our method achieves a slight improvement, whereas for *Eikonal* and *Nonlinear diffusion*, it performs slightly worse. This may be because, with sufficiently rich data, the standard NO can already utilize the data effectively to achieve good performance, leaving little room for the added value of the learned physics to further improve the results. In such cases, the benefit of incorporating physics knowledge becomes

Table 6: The relative $L_2$ error with using system-identification–then–neural-operator on *Darcy flow* and *Nonlinear Diffusion* benchmarks.

(a) Predicting $u$ via different architectures on *Darcy flow*.

| Training size | 5 | 10 | 20 | 30 |
|---|---|---|---|---|
| SINDy → FNO | 0.4624±0.0147 | 0.3382±0.0113 | 0.2725±0.0115 | 0.2222±0.0033 |
| $\phi$-network → FNO | 0.4492±0.0196 | 0.2638±0.0132 | 0.1646±0.0168 | 0.1013±0.0023 |
| FNO | 0.4915±0.0210 | 0.3870±0.0118 | 0.2783±0.0212 | 0.1645±0.0071 |
| Ours | **0.1716±0.0048** | **0.0956±0.0084** | **0.0680±0.0031** | **0.0642±0.0010** |

(b) Predicting $u$ via different architectures on *Nonlinear Diffusion*.

| Training size | 5 | 10 | 20 | 30 |
|---|---|---|---|---|
| SINDy → FNO | 0.2703±0.0048 | 0.2400±0.0059 | 0.1908±0.0123 | 0.1213±0.0034 |
| $\phi$-network → FNO | 0.1603±0.0109 | 0.0943±0.0037 | 0.0540±0.0043 | 0.0361±0.0013 |
| FNO | 0.2004±0.0083 | 0.1242±0.0046 | 0.0876±0.0061 | 0.0551±0.0021 |
| Ours | **0.0105±0.0016** | **0.0066±0.00023** | **0.0049±0.00037** | **0.0038±0.00039** |

marginal. Furthermore, our alternating updating mechanism brings additional optimization workload, which may introduce additional complexity to the learning process.

**Comparison with NO training using ground-truth physics.** Our work is motivated by scenarios where the underlying physics is unknown. It is therefore insightful to compare our approach with training NO using the ground-truth physics, as done in PINO. This comparison allows us to examine how our learned "pseudo" physics differs from the true physics in facilitating operator learning. To this end, we tested PINO on the Poisson and Advection cases, using finite difference approximations to compute the derivatives. We used FNO as the baseline. The PDE residual was incorporated into the training loss. As shown in Table 5, while PINO consistently achieves even better results, the performance of our method is close to PINO's. This demonstrates that our learned black-box representation, despite lacking interpretability, can provide a similar benefit in improving operator learning performance.

**Learning behavior.** We examined the learning behavior of our method, which conducts an iterative, alternatingly fine-tuning process. We employed one *Darcy Flow*, one *nonlinear diffusion* and one *Eikonal* dataset, each with 30 examples. We show the test relative $L_2$ error along with the iterations in Fig. 8. As we can see, the predictive performance of our algorithm kept improving and tended to converge at last, affirming the efficacy of our learning process.

**Ablation study on the weight $\lambda$.** We examined the effect of the weight $\lambda$ of our "pseudo physics"; see equation 5. To this end, we employed *Darcy Flow*, *nonlinear diffusion*, and *Eikonal*, each with 30 examples for training. We varied $\lambda$ from $[0.5, 10^2]$, and run PPI-FNO and PPI-DONet on these datasets. From Fig. 9, we can see that across a wide range of $\lambda$ values, PPI-FNO and PPI-DONet can consistently outperform the standard FNO and DONet respectively by a large margin. However, the choice of $\lambda$ does have a significant influence on the operator learning performance, and the best choice is often in between.

**Ablation study on pseudo physics network $\phi$.** To confirm the efficacy of our designed network $\phi$ in facilitating operator learning, we considered alternative designs for $\phi$: (1) using standard FNO to predict $f$ directly from $u$; no derivative information is included in the input; (2) removing the convolution layer in our model, and just keeping the fully-connected layers, namely MLP; the derivative information is not included in the input. With different designs of $\phi$, we evaluated the PPI learning performance on the Darcy Flow benchmark. The relative $L_2$ errors in predicting $f$ via $\phi$ and predicting $u$ are reported in Table 7. It can be seen that our design of $\phi$ consistently outperforms alternative architectures by a notable margin, showing the effectiveness of learning a (black-box) PDE representation and improving the operator learning.

It worth noting that the PDE network $\phi$ is trained on exactly the same input distribution as the main operator network and all the baselines. We neither extrapolate beyond the original spatial domain nor alter the distribution over coefficients or initial conditions. When we refer to "randomly sampled inputs", these are drawn from the same domain and input distribution as the training data and are used solely as additional collocation points for evaluating the pseudo-PDE residual, rather than as new labeled pairs. In particular,

since only input functions $f'$ are sampled, they cannot be used as $(f, u)$ pairs in the original neural-operator training.

Table 7 also includes a variant in which we simply employ an FNO as a drop-in replacement for the $\phi$-network, trained under the same input distribution. This variant is consistently much worse than our design, indicating that the performance gains do not stem from providing the PDE network with a larger or richer input distribution, nor from increased model capacity alone, but rather from the specific PDE-structured regularization induced by our $\phi$-network.

Table 7: The relative $L_2$ error with using different architectures of $\phi$ in pseudo-physics-informed (PPI) learning on *Darcy flow* benchmark.

(a) Predicting $f$ via $\phi$ with different architectures.

| Training size | 5 | 10 | 20 | 30 |
|---|---|---|---|---|
| FNO | 0.7229±0.0318 | 0.5759± 0.0126 | 0.4257± 0.0106 | 0.3160± 0.0037 |
| MLP | 0.7169±0.0160 | 0.6598± 0.0056 | 0.6464± 0.0029 | 0.6277± 0.0032 |
| Ours | **0.2285 ± 0.0147** | **0.1392 ± 0.0080** | **0.0898 ± 0.0046** | **0.0688 ± 0.0032** |

(b) Predicting $u$.

| Training size | 5 | 10 | 20 | 30 |
|---|---|---|---|---|
| PPI-FNO with FNO as $\phi$ | 0.5853±0.0153 | 0.3871± 0.0124 | 0.2613± 0.0190 | 0.1629± 0.0064 |
| PPI-FNO with MLP as $\phi$ | 0.7262±0.0920 | 0.5516± 0.0699 | 0.4568± 0.0857 | 0.3983± 0.1051 |
| Standard FNO | 0.4915 ± 0.0210 | 0.3870 ± 0.0118 | 0.2783 ± 0.0212 | 0.1645 ± 0.0071 |
| Ours | **0.1716 ± 0.0048** | **0.0956 ± 0.0084** | **0.0680 ± 0.0031** | **0.0642 ± 0.0010** |

Table 8: The relative $L_2$ error of PPI learning by incorporating different orders of derivatives. During the comparison with other operator learning methods, we used derivative orders up to 2 to run our method.

(a) Predicting $f$ via $\phi$.

| Training size | 5 | 10 | 20 | 30 |
|---|---|---|---|---|
| order 0 | 0.7126±0.0131 | 0.5733±0.0208 | 0.4812±0.0399 | 0.3445±0.0182 |
| order $\leq 1$ | 0.2926±0.0118 | 0.2006±0.0047 | 0.1379±0.0051 | 0.1084±0.0053 |
| order $\leq 2$ | 0.2285±0.0147 | 0.1392±0.0080 | 0.0898±0.0046 | 0.0688±0.0032 |
| order $\leq 3$ | **0.2058±0.0192** | **0.1123±0.0039** | **0.0712±0.0021** | **0.0585±0.0030** |

(b) Predicting $u$.

| Training size | 5 | 10 | 20 | 30 |
|---|---|---|---|---|
| order 0 | 0.6352±0.0673 | 0.4523±0.0621 | 0.3570±0.0658 | 0.2737±0.0643 |
| order $\leq 1$ | 0.3386±0.0259 | 0.2161±0.0083 | 0.1645±0.0114 | 0.1197±0.0132 |
| order $\leq 2$ | **0.1716±0.0048** | **0.0956±0.0084** | **0.0680±0.0031** | **0.0642±0.0010** |
| order $\leq 3$ | 0.2959±0.0381 | 0.1719±0.0213 | 0.1193±0.0158 | 0.0828±0.0054 |

Table 9: The relative $L_2$ error with using different distribution in $\phi$ network learning on *Nonlinear Diffusion* and *Advection* benchmark.

(a) testing $f$ via $\phi$ on different distribution of Nonlinear Diffusion benchmark.

| Training size | 5 | 10 | 20 | 30 |
|---|---|---|---|---|
| $\phi$-network test out of distribution | 0.0092±0.0006 | 0.0071±0.0002 | 0.0060±0.0003 | 0.0047±0.0005 |
| $\phi$-network test in distribution | 0.0047±0.0004 | 0.0034±0.0004 | 0.0028±0.0002 | 0.0023±0.0003 |

(b) testing $f$ via $\phi$ on different distribution of Advection benchmark.

| Training size | 20 | 30 | 50 | 80 |
|---|---|---|---|---|
| $\phi$-network test out of distribution | 0.0556±0.0077 | 0.0485±0.0072 | 0.0461±0.0057 | 0.0432±0.0059 |
| $\phi$-network test in distribution | 0.0467±0.0065 | 0.0423±0.0049 | 0.0397±0.0033 | 0.0365±0.0043 |

**Ablation study on the choice of derivatives.** We further investigated the PPI learning performance with respect to the choice of derivatives used in our pseudo physics network. Specifically, we tested PPI-FNO

Table 10: The relative $L_2$ error with using different distribution in pseudo-physics-informed (PPI) learning on *Nonlinear Diffusion* and *Advection* benchmark.

(a) Predict $u$ on different distribution of Nonlinear Diffusion benchmark.

| *Training size* | 5 | 10 | 20 | 30 |
|---|---|---|---|---|
| FNO | 0.1669±0.0084 | 0.1009±0.0012 | 0.0837±0.0016 | 0.0626±0.0009 |
| Ours | **0.0229 ± 0.0088** | **0.0179 ± 0.0013** | **0.0135 ± 0.0010** | **0.0103 ± 0.0003** |

(b) Predict $u$ on different distribution of Advection benchmark.

| *Training size* | 20 | 30 | 50 | 80 |
|---|---|---|---|---|
| FNO | 0.7339±0.0176 | 0.7086±0.0076 | 0.6838±0.0258 | 0.6484±0.0027 |
| Ours | **0.6175 ± 0.0354** | **0.6099 ± 0.0227** | **0.6231 ± 0.0206** | **0.6471 ± 0.0148** |

Table 11: Parameter counts for FNO and DONet with PPI variations across different problems. The training size is 30.

| Parameter count | FNO | PPI-FNO (increase) | DONet | PPI-DONet (increase) |
|---|---|---|---|---|
| Darcy-flow | 1,188,353 | 1,229,476 (+3.46%) | 2,084,704 | 2,125,827 (+1.97%) |
| Nonlinear-diffusion | 1,188,353 | 1,197,220 (+0.75%) | 824,501 | 833,368 (+1.08%) |
| Eikonal | 1,188,353 | 1,197,220 (+0.75%) | 824,501 | 833,368 (+1.08%) |
| Poisson | 1,188,353 | 1,197,220 (+0.75%) | 824,501 | 833,368 (+1.08%) |
| Advection | 1,188,353 | 1,197,220 (+0.75%) | 210,101 | 218,968 (+4.22%) |

on the Darcy-flow benchmark and varied the order of derivatives up to 0, 1, 2, and 3. The performance is reported in Table 8. We can see that although the accuracy of $\phi$ with derivatives up to the third order is slightly better than with derivatives up to the second order, the best operator learning performance was still achieved using derivatives up to the second order (which was used in our evaluations). This might be because higher-order derivative information can cause overfitting in the pseudo physics network $\phi$ to a certain degree. Such higher-order information may not be critical to the actual mechanism of the physical system and can therefore impede the improvement of operator learning performance. Nevertheless, the framework itself is not fundamentally limited to second-order terms: for higher-order PDEs, the input set of the surrogate $\phi$-network can be straightforwardly extended to include third- or higher-order derivatives without modifying the core algorithm.

**Ablation study on the out-of-distribution (OOD) robustness.** Beyond in-distribution accuracy, it is also important to understand how our method behaves under distribution shifts. We therefore investigate OOD robustness at two levels: (i) the pseudo physics network $\phi$, and (ii) the end-to-end PPI-FNO (comparison to FNO under identical OOD splits and training budgets). We construct OOD test sets by shifting the input distribution: for Advection we use squared–exponential (SE) Gaussian kernels with test length scale 0.15 (training 0.25); for Nonlinear Diffusion we use test length scale 0.10 (training 0.20).

Rather than interpreting internal weights, we validate the learned surrogate by its input–output behavior. We compare $\phi$'s predicted dynamic responses against ground truth under challenging initial conditions for both in-distribution (ID) and OOD test fields, and we quantify alignment with physical principles. Across training sizes (Advection: $N \in \{20, 30, 50, 80\}$; Nonlinear Diffusion: $N \in \{5, 10, 20, 30\}$), $\phi$ remains closely aligned with the reference solutions on ID data and shows robust performance under moderate OOD shifts reported in Table 9 and heatmaps Figure 10 and Figure 11, These observations indicate that the surrogate captures physically plausible behavior even as a black box.

We then evaluate PPI-FNO and FNO on the OOD splits with matched optimization settings. On Nonlinear Diffusion, both methods experience only mild degradation relative to ID, while PPI-FNO consistently outperforms FNO across $N$. On Advection, a larger length–scale shift induces substantial degradation for all models; PPI-FNO remains competitive and typically stronger than FNO, but—as expected for operator extrapolation—cannot fully recover accuracy far outside the training range. The performance is reported in Table 10. Overall, these results suggest that coupling with $\phi$ improves sample efficiency and robustness under moderate shifts, but does not eliminate the fundamental difficulty of strong-range extrapolation.

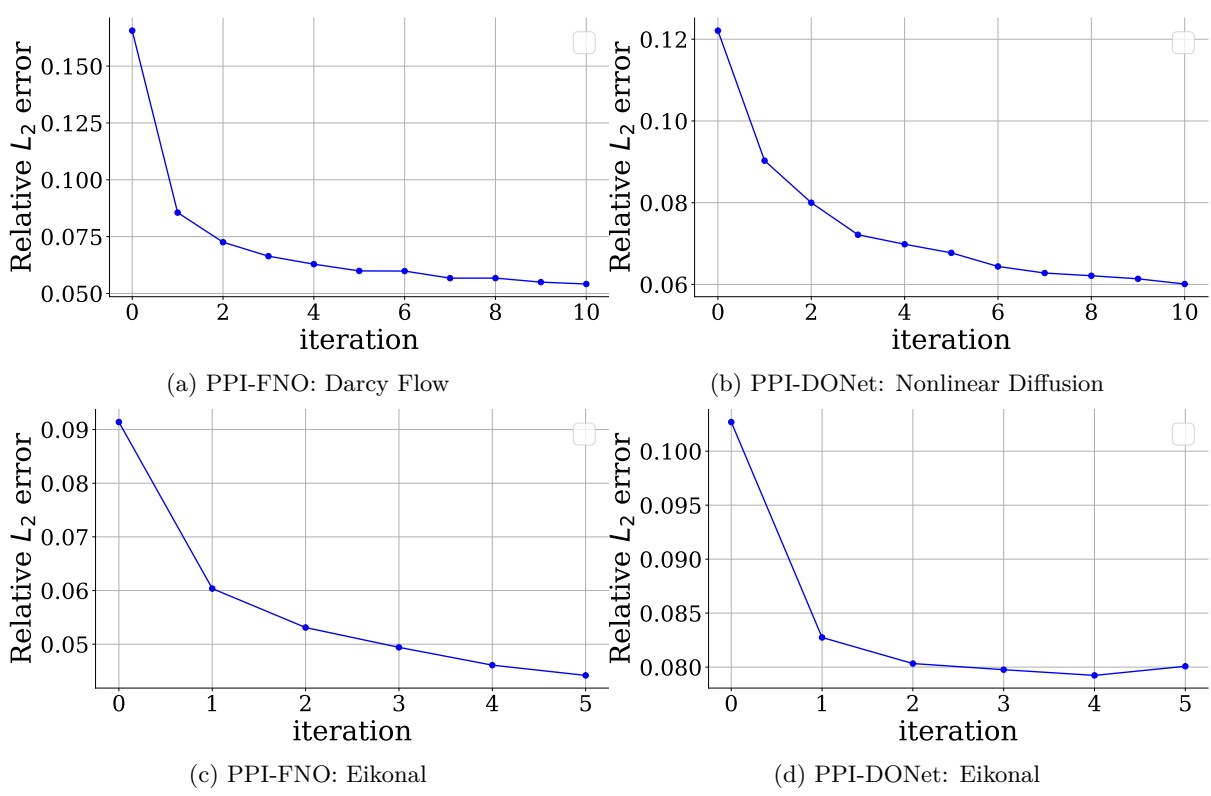

Figure 8: Predictive performance *vs.* alternatingly fine-tuning iterations.

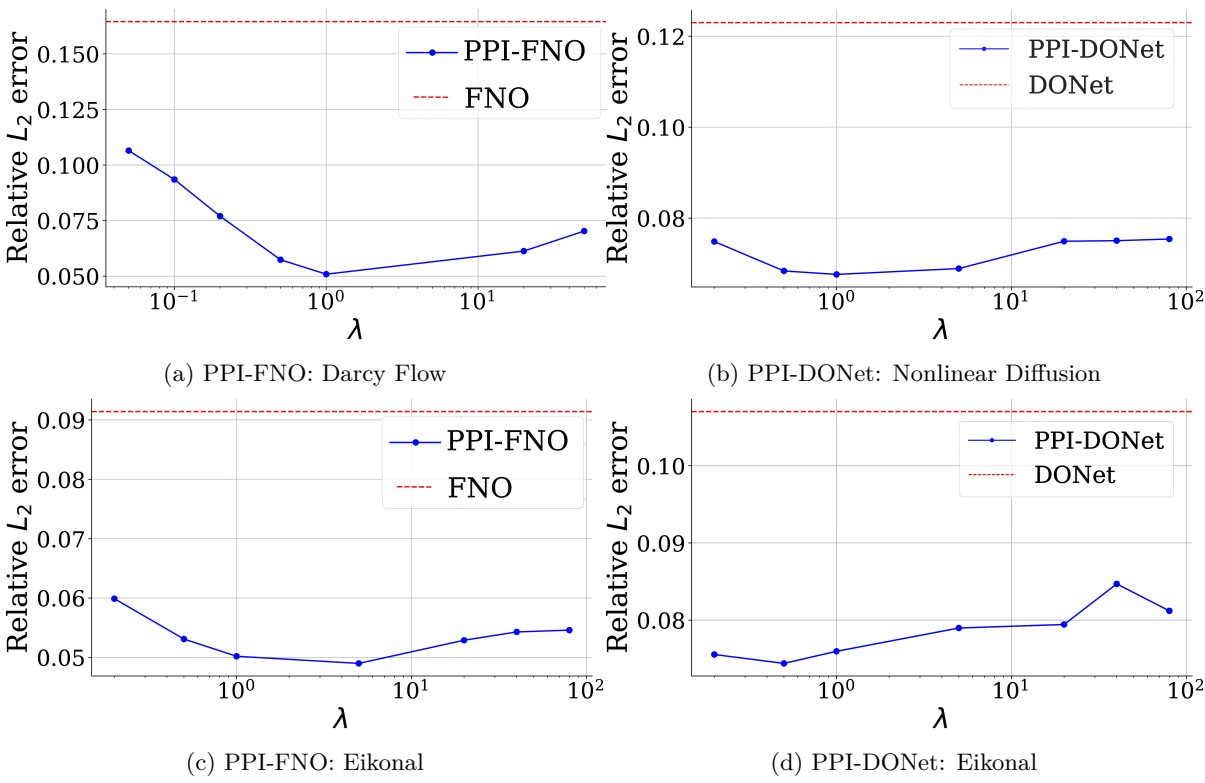

Figure 9: Predictive performance *vs.* weight $\lambda$.

Table 12: running time cost of FNO and PPI-FNO on *darcy-flow* and *Poisson* benchmark.

(a) running time cost of standard FNO

| *Training size* | 5 | 10 | 20 | 30 |
|---|---|---|---|---|
| darcy-flow | 14 | 14 | 14 | 15 |
| poisson | 13 | 13 | 14 | 14 |

(b) running time cost of PPI-FNO

| *Training size* | 5 | 10 | 20 | 30 |
|---|---|---|---|---|
| darcy-flow | 984 | 1001 | 1036 | 1064 |
| poisson | 869 | 881 | 908 | 935 |

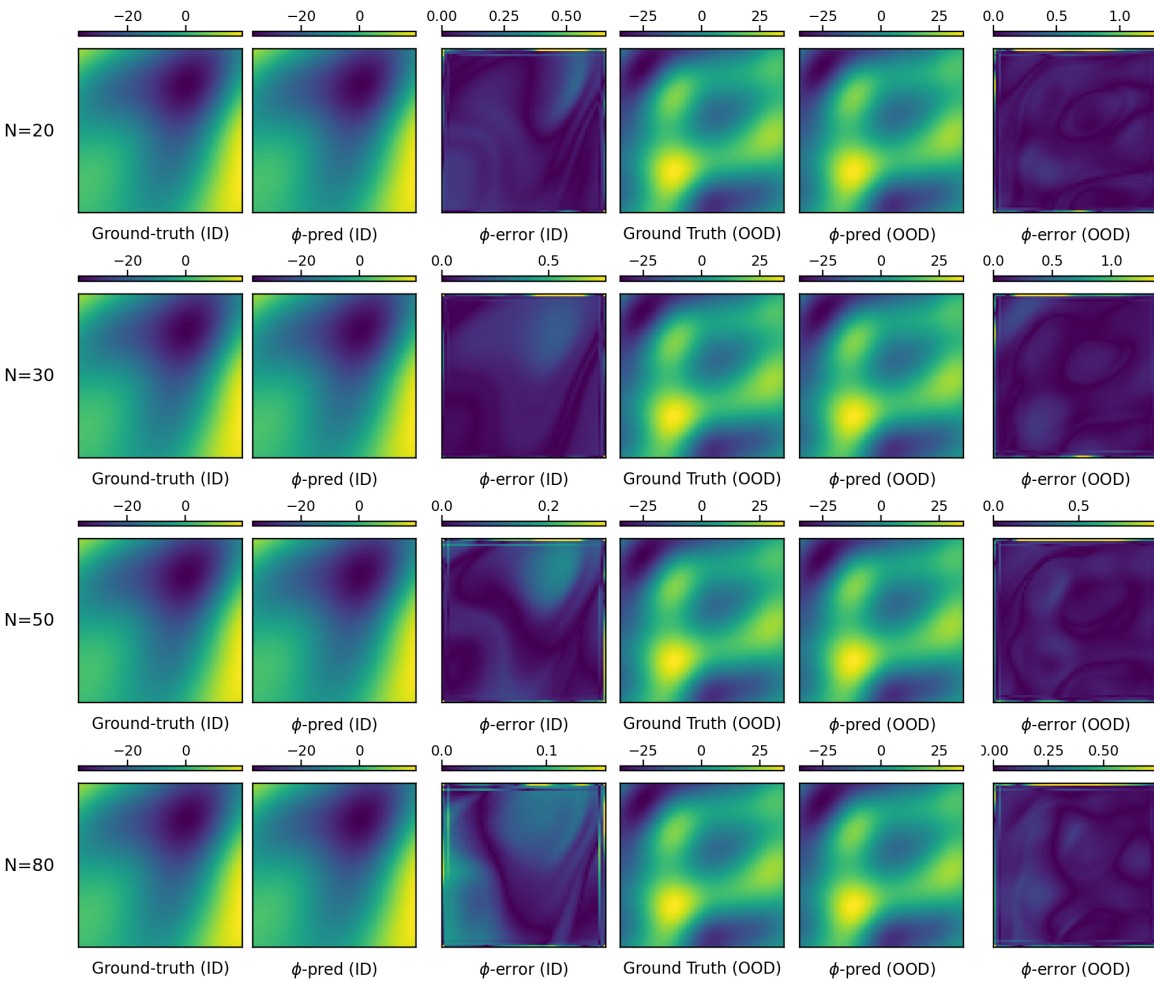

Figure 10: Advection heatmap (ID / OOD).

**Ablation study on noisy data.** The main experiments in this work focus on the challenge of limited data availability, i.e., a small number of solution snapshots $N$, rather than low spatial resolution. The use of high-resolution (128×128), noise-free data in our benchmarks was intentional: it provides a controlled and stringent testbed to evaluate our method's ability to learn a generalizable physical model from few examples. If the data were coarse or noisy, it would be difficult to disentangle the effect of data quality from the method's intrinsic performance. In addition, high-fidelity data is typically expensive and thus limited

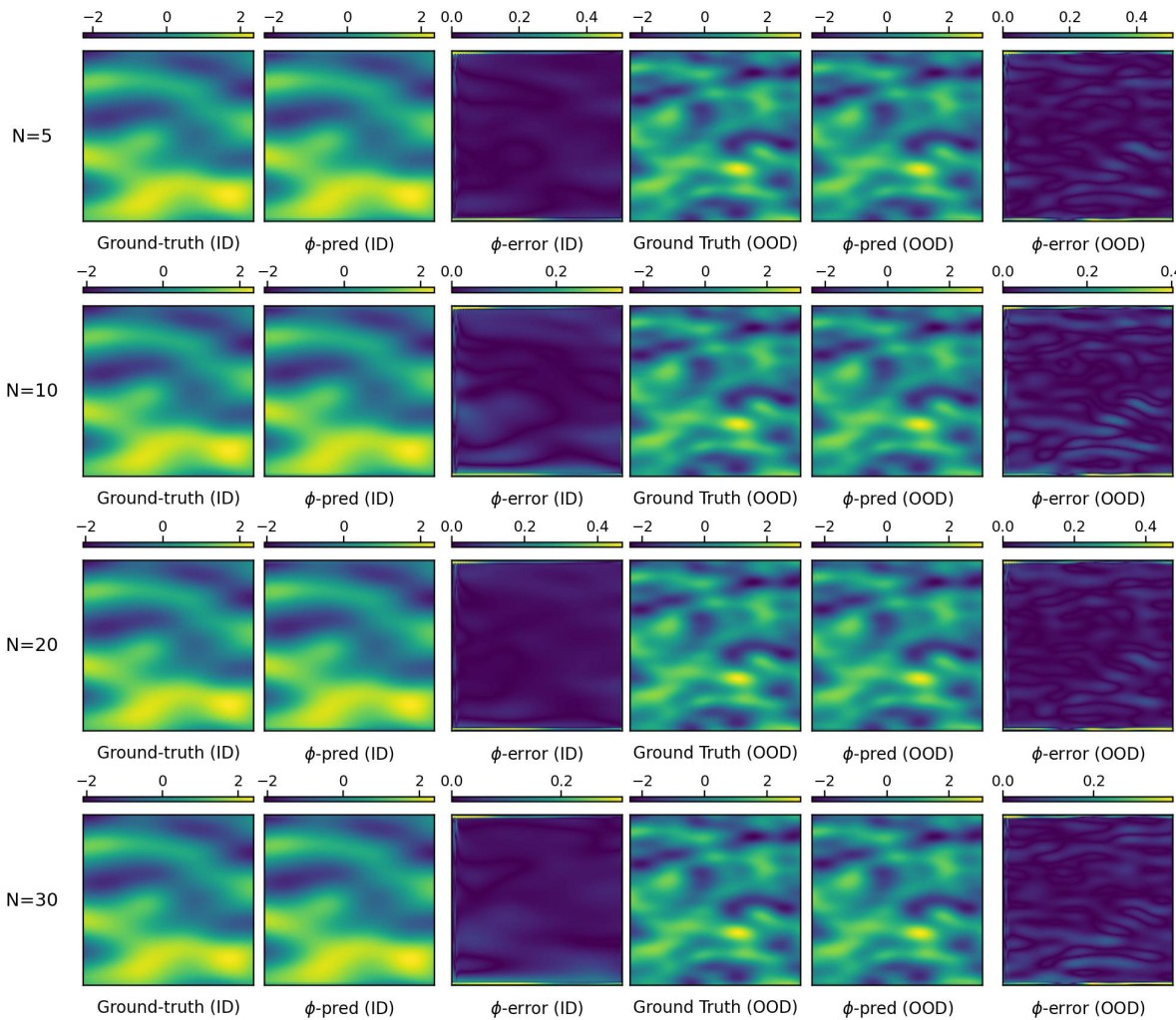

Figure 11: Nonlinear diffusion heatmap (ID / OOD).

in quantity, whereas low-fidelity data is easy to acquire and rarely scarce. Hence, our setting—with few high-fidelity samples—accurately reflects the intended use case of the proposed framework.

Nonetheless, we evaluated the robustness of our framework to data sparsity and noise using the nonlinear diffusion benchmark. Each training input–output pair, originally sampled on a $128 \times 128$ noise-free grid, was randomly subsampled to 400 function values, to which we added 5% Gaussian noise. These noisy samples were then denoised and interpolated back to the $128 \times 128$ grid using Gaussian process interpolation (since FNO requires gridded data). We subsequently trained our PPI-NO model under this setting.

As shown in Table 13, our method consistently outperforms FNO across all training sizes, even in the presence of noise, though with the expected performance degradation compared to the clean-data experiments.

**Memory and Time Cost.** Our PPI-NO is more costly than standard NO since we need to train an additional "pseudo" physics network $\phi$ along with the NO model. However, the network $\phi$ is small as compared to the NO component — $\phi$ is simply a pixel-wise MLP coupled with one convolution filter, resulting in a marginal increase in memory cost. Table 11 shows the parameter count of FNO, DONet and their pseudo-physics-informed versions. On average, PPI-FNO increases the number of parameters over FNO by 1.29% while PPI-DONet over DONet by 1.89%. In terms of training time, the alternating

Table 13: The relative $L_2$ error training with noise data in pseudo-physics-informed (PPI) learning on *Nonlinear Diffusion* benchmark.

| Training size | 5 | 10 | 20 | 30 |
|---|---|---|---|---|
| FNO | 0.2037±0.0114 | 0.1217±0.0056 | 0.0891±0.0060 | 0.0606±0.0020 |
| Ours | **0.1465±0.0062** | **0.0848±0.0041** | **0.0593±0.0023** | **0.0504±0.0026** |

optimization between the solution operator and the pseudo-physics surrogate is by design: it allows the two components to co-evolve, but introduces extra wall-clock time relative to single-network baselines. The cost grows approximately linearly with the number of alternations (and the per-iteration epoch budget). In return, we observe consistent accuracy gains in the low-data regime. To make the computational overhead transparent, we report end-to-end wall-clock times on the Darcy flow and Poisson benchmarks in Table 12; all measurements are obtained on a Linux workstation equipped with an NVIDIA GeForce RTX 4090 (24 GB).

# 6 Discussions and Limitations

## 6.1 More general PDE representation

Our framework can extend to more general settings where operator learning involves mapping from coefficients and/or initial and/or boundary conditions to solutions.

**Coefficient input.** When $f$ serves as a coefficient function in the PDE, our $\phi$-network can be trained to learn equations of the form

$$L(x, u(x), S_1(u)(x), \ldots, S_Q(u)(x), f(x), S_1(f)(x), \ldots, S_Q(f)(x)) = 0,$$

where $S_i$ denotes derivative operators, e.g., $u_t + \nabla(f(x) \cdot u(x)) = u_{xx}$. The training loss can be constructed analogously, ensuring that the $\phi$-network's predictions approach zero on valid training examples. In this setting, a key challenge is the identifiability of the learned pseudo-PDE. For example, the network can collapse to a trivial zero output for any input. To address the challenge, we can employ more informative training objectives.

First, we can design the PDE loss so that the residual is close to zero on correctly paired coefficient–solution samples but large on randomly mismatched pairs drawn from the same coefficient distribution. For such correctly paired data $(u, f)$, we encourage a zero residual, where the $L^2(\Omega)$ norm is approximated by averaging over randomly sampled locations $x$ in the domain $\Omega$. In contrast, if we keep the same solution $u$ but pair it with a coefficient $\tilde{f}$ randomly drawn from the same coefficient distribution, we expect that

$$L^2(x, u(x), S_1(u)(x), \ldots, S_Q(u)(x), \tilde{f}(x), S_1(\tilde{f})(x), \ldots, S_Q(\tilde{f})(x)) > 0,$$

This suggests a contrastive loss:

$$\mathcal{J}(\theta) = \|L(u, f)\|^2_{L^2(\Omega)} - \|L(u, \tilde{f})\|^2_{L^2(\Omega)},$$

where each $L^2(\Omega)$ norm is approximated by averaging the squared residual over randomly sampled locations in the same domain $\Omega$ as the data. The first term enforces a small residual on true coefficient–solution pairs, while the second term explicitly penalizes zero (or very small) residuals on randomly mismatched pairs. Under this objective, a constant-zero PDE network is no longer optimal: the model can strictly decrease $\mathcal{J}(\theta)$ by keeping the residual small on true pairs but making it large on random mismatches, which reduces degeneracy and improves identifiability in practice.

Second, we can also use likelihood-based generative models such as PixelCNN (van den Oord et al., 2016), learning to maximize the probability density of coefficients and solutions at the sampled locations. The probability density is parameterized by the network $\phi$. This is complementary to the residual-based objective above.

**Initial conditions.** For cases involving mappings from initial conditions to solutions, the PDE can be written as

$$\frac{\partial u}{\partial t} = g(u, t, x, S_1(u), \ldots, S_Q(u)).$$

When trajectory data are available, $\frac{\partial u}{\partial t}$ can be estimated directly from data, and $g$ can then be learned via the $\phi$-network. The mapping from $f$ to $u$ at a target time $T$ can be represented as $u_T = f + \int_0^T g \, dt$, where numerical integration provides an approximate PDE representation.

**Boundary conditions.** When $f$ represents Dirichlet boundary conditions, the PDE surrogate can be modeled as

$$u(x) = f(x) + \alpha(x) \, \phi(x, u, S_1(u), \ldots, S_Q(u)),$$

where $f$ enforces the boundary condition and $\alpha(x)$ is a composite distance function that vanishes on the boundaries. Incorporating more general boundary conditions into the PDE surrogate learning, however, remains an open research question.

## 6.2 Limitations and future work

We acknowledge that our Pseudo Physics-Informed Neural Operator (PPI-NO) learning framework has several limitations.

First, our empirical validation concentrates on standard operator-learning benchmarks and does not cover highly nonlinear, multi-scale PDEs (e.g., Navier–Stokes, Euler). These regimes exhibit strong stiffness, intermittent structures, and scale separation that can destabilize training, alter inductive biases, and invalidate assumptions implicit in our datasets. As a result, the reported gains should not be interpreted as evidence of effectiveness in these harder settings. The methodology, loss balancing, and data priors used here may interact unfavorably with multi-scale dynamics, making generalization uncertain even when numerical resolution is increased.

Second, because the surrogate physics network $\phi$ is trained from limited examples, it can overfit spurious regularities in the training distribution and internalize artifacts of discretization or preprocessing. Such overfitting may not be apparent from aggregate errors yet can surface as unstable extrapolation, biased fluxes, or residual patterns that mimic physics without reflecting underlying conservation or constitutive structure. The risk grows when the training inputs have narrow support (e.g., restricted length scales or amplitudes), when measurement noise is structured, or when the training loss inadvertently rewards easy-to-fit nonphysical correlations. In these cases, $\phi$ can act as a powerful but misleading prior on the operator learner, tightening training loss while harming reliability, like we showed in the ablation study on the choice of derivatives.

Third, the framework validates $\phi$ primarily by its input–output behavior, not by interpretability of its internal form. Consequently, apparent agreement with test data can be driven by correlations specific to the sampling protocol, interpolation scheme, or solver discretization, rather than by genuine physical invariants. This ambiguity is amplified under PDE range shift: improvements in in-distribution metrics do not guarantee preservation of causal structure when inputs depart from the training manifold. Without direct identifiability of operators or invariants, it is difficult to determine when $\phi$ captures a true governing relation versus a dataset-dependent proxy. Hence, conclusions about "learned physics" must be treated cautiously, especially when the ground-truth PDE is unknown.

Fourth, grid-based function values are strongly correlated, so the effective sample size for training the surrogate physics network is much smaller than the raw number of grid points. Although our design uses convolutional layers to aggregate local neighborhoods—thereby exploiting spatial correlation and improving representation quality relative to a pointwise MLP, this aggregation does not create independent evidence. In practice, it can lead to underestimated uncertainty and overconfident losses when many nearby pixels convey redundant

information. Moreover, the degree and structure of correlation are resolution-dependent: a fixed receptive field in pixels corresponds to different physical extents across meshes, and discretization choices (stencil width, padding, interpolation) imprint resolution-specific artifacts. Under resolution shifts—or for PDEs with longer correlation lengths, anisotropy, or nonlocal couplings—these effects can induce systematic bias in the learned surrogate, degrade cross-resolution generalization, and blur the distinction between genuine physical dependencies and grid-induced regularities. Consequently, improvements observed at a single resolution should not be interpreted as evidence of resolution invariance.

Finally, $\phi$ trades interpretability for predictive flexibility. This black-box character constrains the framework's role in scientific workflows that require explicit, human-readable laws, term-level attribution, or formal guarantees. Even when predictions are accurate, the absence of a transparent structure impedes scrutiny, reproducibility across laboratories with different preprocessing pipelines, and integration with analytical theory or established numerical methods. Moreover, the coupling between $\phi$ and the operator learner can obscure the source of errors: performance degradations may arise from $\phi$, the operator model, or their interaction, complicating diagnosis. As a result, the method is better viewed as a predictive surrogate within well-specified data regimes than as a tool for definitive law discovery.

## 7 Conclusion

We have presented a Pseudo Physics-Informed Neural Operator (PPI-NO) learning framework. PPI-NO is based on our observation that a PDE system is often characterized by a *local* combination of the solution and its derivatives. This property makes it feasible to learn an effective representation of the PDE system, even with limited data. While the physics delineated by PPI-NO might not precisely reflect true physical phenomena, our findings reveal that this method significantly enhances the efficiency of operator learning with limited data quantity.

However, our current method cannot learn PDE representations for which the input function $f$ is the *initial condition*. In such cases, the mapping from the solution function to the initial condition requires a reversed integration over time, hence we cannot decouple the derivatives. To address this problem, we plan to explicitly model the temporal dependencies in the PDE representation, such as via the neural ODE design (Chen et al., 2018).

## 8 Acknowledgements

This work has been supported by MURI AFOSR grant FA9550-20-1-0358 (Machine Learning and Physics-Based Modeling and Simulation), NSF CAREER Award IIS-2046295, NSF CSSI-2311685, DARPA SURGE HR0011-25-C-0036 (Rapid certification of additive manufactured components using ML/AI and physics-based modeling), and NSF DMS-2529112 (Rapid Digital Twin model updating).

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

# Appendix

## A    Experimental Details

### A.1    Darcy Flow

We considered a steady-state 2D Darcy Flow equation (Li et al., 2020b),

$$-\nabla \cdot (a(x)\nabla u(x)) = f(x) \ \ x \in (0,1)^2,$$
$$u(x) = 0 \ \ x \in \partial(0,1)^2, \tag{6}$$

where $u(\mathbf{x})$ is the velocity of the flow, $a(\mathbf{x})$ characterizes the conductivity of the media, and $f(\mathbf{x})$ is the source function that can represent flow sources or sinks within the domain. In the experiment, our goal is to predict the solution $u$ given the external source $f$. To this end, we fixed the conductivity $a$, which is generated by first sampling a Gauss random field $\alpha$ in the domain and then applying a thresholding rule: $a(\mathbf{x}) = 4$ if $\alpha(\mathbf{x}) < 0$, otherwise $a(\mathbf{x}) = 12$. We then used another Gauss random field to generate samples of $f$. We followed (Li et al., 2020b) to solve the PDE using a second-order finite difference solver and collected the source and solution at a $128 \times 128$ grid.

### A.2    Nonlinear Diffusion PDE

We next considered a nonlinear diffusion PDE,

$$\partial_t u(x,t) = 10^{-2}\partial_{xx}u(x,t) + 10^{-2}u^2(x,t) + f(x,t),$$
$$u(-1,t) = u(1,t) = 0, \ \ u(x,0) = 0, \tag{7}$$

where $(x,t) \in [-1,1] \times [0,1]$. Our objective is to predict the solution function $u$ given the source function $f$. We used the solver provided in (Lu et al., 2022), and discretized both the input and output functions at a $128 \times 128$ grid. The source $f$ was sampled from a Gaussian process with an isotropic square exponential (SE) kernel for which the length scale was set to 0.2.

### A.3    Eikonal Equation

Third, we employed the Eikonal equation, widely used in geometric optics and wave modeling. It describes given a wave source, the propagation of wavefront across the given media where the wave speed can vary at different locations. The equation is as follows,

$$|\nabla u(\mathbf{x})| = \frac{1}{f(\mathbf{x})}, \ \ \mathbf{x} \in [0,256] \times [0,256], \tag{8}$$

where $u(\mathbf{x})$ is the travel time of the wavefront from the source to location $\mathbf{x}$, $|\cdot|$ denotes the Euclidean norm, and $f(\mathbf{x}) > 0$ is the speed of the wave at $\mathbf{x}$.

In the experiment, we set the wave source at $(0,10)$. The goal is to predict the travel time $u$ given the heterogeneous wave speed $f$. We sampled an instance of $f$ using the expression:

$$f(\mathbf{x}) = \max(g(\mathbf{x}), 0) + 1.0,$$

where $g(\cdot)$ is sampled from a Gaussian process using the isotropic SE kernel with length-scale 0.1. We employed the `eikonalfm` library (https://github.com/kevinganster/eikonalfm/tree/master) that implements the Fast Marching method Sethian (1999) to compute the solution $u$.

### A.4    Poisson Equation

Fourth, we considered a 2D Poisson Equation,

$$-\Delta u = f, \ \ \text{in } \Omega = [0,1]^2, \ \ u|_{\partial D} = 0. \tag{9}$$

where $\Delta$ is the Laplace operator. The solution is designed to take the form, $u(x_1, x_2) = \frac{1}{\pi K^2} \sum_{i=1}^{K} \sum_{j=1}^{K} a_{ij}(i^2 + j^2)^r \sin(i\pi x_1) \cos(j\pi x_2)$, and $f(x_1, x_2)$ is correspondingly computed via the equation. To generate the dataset, we set $K = 5$ and $r = 0.5$, and independently sampled each element $a_{ij}$ from a uniform distribution on $[0, 1]$.

### A.5 Advection Equation

Fifth, we considered a wave advection equation,

$$\frac{\partial u}{\partial t} + \frac{\partial u}{\partial x} = f, \quad x \in [0, 1], \quad t \in [0, 1]. \tag{10}$$

The solution is represented by a kernel regressor, $u(\mathbf{x}) = \sum_{j=1}^{M} w_j k(\mathbf{x}, \mathbf{z}_j)$, and the source $f$ is computed via the equation. To collect instances of $(f, u)$, we used the square exponential (SE) kernel with length-scale 0.25. We randomly sampled the locations $\mathbf{z}_j$ from the domain and the weights $w_j$ from a standard normal distribution.

### A.6 Fatigue Modeling

We considered predicting the SIF values along semi-elliptic surface cracks on plates, as shown in Fig 3. The SIF value can be viewed as a function of the angle $\phi \in [0, \pi]$, which decides the location of each point on the crack surface. The geometry parameters that characterize the crack shape and position were used as the input, including $a/c$, $a/t$ and $c/b$. In the operator learning framework, the input can be viewed as a function with three constant outputs. The dataset was produced via a high-fidelity FE model under Mode I tension (Merrell et al., 2024). Each data instance includes 128 samples of the SIF values drawn uniformly across the range of $\phi$.

