# OpenReview forum: "Pseudo-Physics-Informed Neural Operators: Enhancing Operator Learning from Limited Data"
_TMLR — Accepted by TMLR_

### Review · Reviewer_oRPB · 2025-09-07

**Summary Of Contributions:**

The paper introduces the Pseudo Physics-Informed Neural Operator a novel framework that enhances neural operator learning in data-scarce regimes. The method constructs a surrogate PDE system using finite-difference derivatives and learnable convolutional filters, which is then coupled with existing neural operator architectures (e.g., FNO and DeepONet) through an alternating training scheme.

Some of the contributions include:

- Demonstrating that surrogate, data-driven “pseudo-physics” can serve as an effective regularizer for operator learning without requiring full knowledge of governing equations.

- Comprehensive experiments on benchmark PDE problems (Darcy flow, nonlinear diffusion, Eikonal, Poisson, Advection) as well as a challenging fatigue modeling task, showing substantial performance improvements, particularly with very limited training data.

- Ablation studies that carefully assess the role of the pseudo-physics network, derivative orders, convolutional layers, loss weights, and training strategies.

----

Strengths:

- Novel and creative approach to bridging physics-informed and purely data-driven operator learning.

- Strong empirical validation across diverse PDEs and a real-world fatigue modeling application.

- Clear demonstration that the framework is especially effective in low-data regimes.

- Well-written and thorough, with multiple ablations that strengthen the claims.

Weaknesses:

- Limited discussion on how discovered PDEs generalize (especially out-of-distribution generalization), particularly when spurious or nonlinear terms are learned.

- Open questions remain about the applicability to highly nonlinear, multi-scale systems (e.g., Navier–Stokes, Euler equations), where physics-informed methods are known to struggle.

- Some assumptions on data independence (e.g., treating grid points as separate samples) could be misleading

**Additional Comments:**

/

**Audience:**

Yes

**Audience Explanation:**

The work is highly relevant to the TMLR community, particularly researchers in scientific machine learning, operator learning, and data-efficient modeling. The introduction of pseudo-physics-informed learning provides both a conceptual contribution and a practical framework that could inspire further research, especially in domains where ground-truth physics is unknown or partially understood. The unusual and useful fatigue modeling benchmark also broadens its potential impact.

**Broader Impact Concerns:**

No major ethical concerns are apparent.

**Claims And Evidence:**

Yes

**Claims Explanation:**

The authors provide extensive experimental results across five benchmark PDE problems and one applied fatigue modeling scenario. Performance comparisons against FNO, DeepONet, and PINO baselines are fair and transparent. Ablation studies on architecture, hyperparameters, and derivative order further strengthen the evidence. While certain theoretical aspects (e.g., robustness to incorrect pseudo-physics) are not fully explored, the empirical evidence convincingly supports the main claims of improved data efficiency and predictive accuracy.

**Requested Changes:**

Critical: Clarify the limitations of the approach in cases where the discovered PDE is incorrect, especially regarding its effect on generalization (both in-distribution and out-of-distribution). While the paper does acknowledge that the learned pseudo-physics may not exactly reflect the true governing equations and presents empirical evidence that operator learning is relatively robust to inaccuracies in the pseudo-physics network, the discussion remains limited. The paper does not fully analyze potential failure cases, for example, when spurious nonlinear terms are introduced, or when the learned PDE fundamentally misrepresents the underlying system. In such cases, one might expect both in-distribution and out-of-distribution generalization to suffer. A more explicit discussion of these limitations would strengthen the paper and help readers understand the boundary conditions under which pseudo-physics-informed methods remain beneficial.

Critical: Provide some discussion of applicability to nonlinear, multi-scale PDEs (e.g., Navier–Stokes, Euler). Even if experiments are not feasible, a short argument about feasibility and challenges would strengthen the paper.

Optional: Elaborate on whether and how parameters, source terms, or boundary/initial conditions could be incorporated into the pseudo-PDE network.

Optional: Revisit the discussion around data independence, while the framework treats all grid points as separate samples, in many cases solutions are parametrized, and thus this assumption may overstate the effective data richness.

---

> ### Author Response · Authors · 2025-10-18
>
> We thank the reviewer for the constructive comments. C: comments; R: response
>
> >C1: Limited discussion on how discovered PDEs generalize (especially out-of-distribution generalization), particularly when spurious or nonlinear terms are learned?
>
> R1:
> Thank you for raising the question of OOD generalization and the impact of potential spurious terms. To assess how the learned surrogate physics behaves beyond the training distribution, we ran diagnostics on two datasets: Advection and Nonlinear Diffusion. For each dataset, we evaluated the learned phi-network on (i) in-distribution (ID) test fields sampled from squared-exponential (SE) Gaussian kernels with length scale 0.25, and (ii) out-of-distribution (OOD) test fields sampled from SE kernels with length scale 0.15. We also provide heatmaps to visualize the surrogate’s behavior.
>
> Advection heatmaps:
> https://www.dropbox.com/scl/fi/a77q6suaec3u2dlrsa2qm/advection_data_phi_network_heatmap.png?rlkey=c9wjvud23jzmsp0yedtoec1lt&st=dn51nz3h&dl=0
> Advection $\phi$-network error table:
> |  Relative L2| Training size=20  |  30 | 50  |  80 |
> |---|---|---|---|---|
> |$\phi$-network test in different distribution| 0.0092(0.0006) | 0.0071(0.0002) | 0.0060(0.0003) | 0.0047(0.0005)|
> |$\phi$-network test in same distribution| 0.0047(0.0004) | 0.0034(0.0004) | 0.0028(0.0002) | 0.0023(0.0003)|
>
> Nonlinear Diffusion heatmaps:
> https://www.dropbox.com/scl/fi/7jjf1wrsj9g3a8g28j4bt/nonlinear_diffusion_data_phi_network_heatmap.png?rlkey=pu1w4hblypzrfq42pxu05eqq5&st=hlox5g7r&dl=0
> Nonlinear Diffusion $\phi$-network error table:
> |  Relative L2| Training size=5  |  10 | 20  |  30 |
> |---|---|---|---|---|
> |$\phi$-network test in different distribution| 0.0556(0.0077) | 0.0485(0.0072) | 0.0461(0.0057) | 0.0432(0.0059)|
> |$\phi$-network test in same distribution| 0.0467(0.0065) | 0.0423(0.0049) | 0.0397(0.0033) | 0.0365(0.0043)|
>
> Across training sizes (Advection: N in {20, 30, 50, 80}; Nonlinear Diffusion: N in {5, 10, 20, 30}), the phi-network remains closely aligned with ground truth under both ID and moderate OOD shifts, indicating robust generalization despite its black-box parameterization.
>
>
> >C2: Open questions remain about the applicability to highly nonlinear, multi-scale systems (e.g., Navier–Stokes, Euler equations), where physics-informed methods are known to struggle.
>
> R2: We appreciate the reviewer’s comment. Our current method, like other operator-learning approaches (e.g., **PINO**), has been validated on standard benchmarks but has not yet been tested on highly nonlinear, multi-scale PDEs such as **Navier–Stokes** or **Euler** equations—settings where physics-informed methods often encounter optimization difficulties and scale-separation challenges. We acknowledge this as a limitation and plan to explore these more complex regimes in future work.
>
> >C3: Some assumptions on data independence (e.g., treating grid points as separate samples) could be misleading
>
> R3: Thanks for the insightful comments. We agree that the function values on the grids are not independent; thus, the **effective sample size (ESS)** for training our $\phi$-network is smaller than the total number of grid points. In fact, our $\phi$-network design explicitly leverages **convolutional layers** to aggregate information from neighboring grid points, capturing spatial correlations and improving representation quality. As shown in the **ablation study (Table 6)**, this design substantially enhances performance compared to a point-wise MLP baseline.
>
> >C4: Clarify the limitations of the approach in cases where the discovered PDE is incorrect, especially regarding its effect on generalization (both in-distribution and out-of-distribution). While the paper does acknowledge that the learned pseudo-physics may not exactly reflect the true governing equations and presents empirical evidence that operator learning is relatively robust to inaccuracies in the pseudo-physics network, the discussion remains limited. The paper does not fully analyze potential failure cases, for example, when spurious nonlinear terms are introduced, or when the learned PDE fundamentally misrepresents the underlying system. In such cases, one might expect both in-distribution and out-of-distribution generalization to suffer. A more explicit discussion of these limitations would strengthen the paper and help readers understand the boundary conditions under which pseudo-physics-informed methods remain beneficial. Provide some discussion of applicability to nonlinear, multi-scale PDEs (e.g., Navier–Stokes, Euler). Even if experiments are not feasible, a short argument about feasibility and challenges would strengthen the paper.
>
> R4: Thanks for your excellent suggestion. We do agree. We have added a detailed discussion of the limitation following your suggestions, please see R1.

---

> > ### Author Response · Authors · 2025-10-22
> >
> > >C1: Critical: Clarify the limitations of the approach in cases where the discovered PDE is incorrect, especially regarding its effect on generalization (both in-distribution and out-of-distribution). While the paper does acknowledge that the learned pseudo-physics may not exactly reflect the true governing equations and presents empirical evidence that operator learning is relatively robust to inaccuracies in the pseudo-physics network, the discussion remains limited. The paper does not fully analyze potential failure cases, for example, when spurious nonlinear terms are introduced, or when the learned PDE fundamentally misrepresents the underlying system. In such cases, one might expect both in-distribution and out-of-distribution generalization to suffer. A more explicit discussion of these limitations would strengthen the paper and help readers understand the boundary conditions under which pseudo-physics-informed methods remain beneficial.
> >
> > Thank you for raising the question of OOD generalization and the impact of potential spurious terms. We responded to the comment in rebuttal **R1**. In the revised manuscript, we have added the related experiment, analysis, and discussion in **section 5.2, page 17 "Ablation study on the out-of-distribution"**, and also discussed limitations in **section 7 on page 21**.
> >
> >
> > >C2: Critical: Provide some discussion of applicability to nonlinear, multi-scale PDEs (e.g., Navier–Stokes, Euler). Even if experiments are not feasible, a short argument about feasibility and challenges would strengthen the paper.
> >
> > We appreciate the reviewer’s comment. We have added the limitation discussion in the updated manuscript in **section 7, page 21**, and also responded in rebuttal **R2**.
> >
> > >C3: Optional: Elaborate on whether and how parameters, source terms, or boundary/initial conditions could be incorporated into the pseudo-PDE network.
> >
> > Thanks for the insightful comments. After answerd in rebuttal **R5**, we have added the explanation and discussion in **section 3.3, page 6, starting from the last few lines** of our new manuscript.
> >
> >
> > >C4: Optional: Revisit the discussion around data independence, while the framework treats all grid points as separate samples, in many cases solutions are parametrized, and thus this assumption may overstate the effective data richness.
> >
> > Thanks for your excellent suggestion. We have made changes to discuss data independence both in **section 3.1(page 5), 8th line, and in section 7(page 21)**.

---

> ### Author Response · Authors · 2025-10-18
>
> >C5:Elaborate on whether and how parameters, source terms, or boundary/initial conditions could be incorporated into the pseudo-PDE network.
>
> R5: Great suggestion. We believe our framework can naturally extend to more general settings where operator learning involves mapping from **parameters and/or initial and/or boundary conditions to solutions**.
>
> Specifically, when $f$ serves as a **parameter function** in the PDE, our $\phi$-network can be trained to learn equations of the form
>
> $$
> L(x, u(x), S_1(u)(x), \ldots, S_Q(u)(x), f(x), S_1(f)(x), \ldots, S_Q(f)(x)) = 0,
> $$
>
> where $S_i$ denotes derivative operators (e.g., $u_t + \nabla(f(x) \cdot u(x)) = u_{xx}$).
> The training loss can be constructed analogously, ensuring that the $\phi$-network’s predictions approach zero on valid training examples.
> However, to prevent the network from collapsing to a trivial zero output, we additionally sample a large number of random input functions and encourage $\phi$ to produce **non-zero predictions** when the PDE constraint is violated.
>
> For cases involving **mappings from initial conditions to solutions**, the PDE can be written as
>
> $$
> \frac{\partial u}{\partial t} = g(u, t, x, S_1(u), \ldots, S_Q(u)).
> $$
>
> When trajectory data are available, $\frac{\partial u}{\partial t}$ can be estimated directly from data, and $g$ can then be learned via the $\phi$-network. The mapping from $f$ to $u$ at a target time $T$ can be represented as
> $u_T = f + \int_{0}^T g \, dt$,
> where numerical integration provides an approximate PDE representation.
>
> When $f$ represents **Dirichlet boundary conditions**, the PDE surrogate can be modeled as
>
> $$
> u(x) = f(x) + \alpha(x) \, \phi(x, u, S_1(u), \ldots, S_Q(u)),
> $$
>
> where $f$ enforces the boundary condition and $\alpha(x)$ is a composite distance function that vanishes on the boundaries.
> Incorporating more general boundary conditions into the PDE surrogate learning, however, remains an **open research question**.

---

### Review · Reviewer_SQyJ · 2025-09-14

**Summary Of Contributions:**

**Summary**

The paper introduces the Pseudo Physics-Informed Neural Operator (PPI-NO) framework, which couples a neural operator (e.g., FNO or DeepONet) with a learned “pseudo-physics” surrogate PDE network. The surrogate network takes solution values and derivative approximations as inputs and is trained jointly with the operator via alternating optimization. The goal is to improve operator learning in scenarios where governing physics are unknown and labeled data are scarce. The paper presents extensive experiments on benchmark operator learning problems and a fatigue modeling application, along with ablation studies.

**Key strengths** include:
- **(1)** The problem of enhancing neural operators when physical laws are unavailable and labeled data are scarce is an important and underexplored research direction.
-  **(2)** The core methodology is clearly presented, and the alternating training scheme is well motivated.
-  **(3)** The paper provides extensive numerical examples, including ablations, which is commendable.

**Key weaknesses** include:
- **(1) Weak motivation**: The central idea is to improve neural operator training by introducing a surrogate “pseudo-physics” network that acts as a learned PDE. However, I find the motivation weak. There already exist advanced system identification methods capable of accurately recovering governing equations from limited data. A more natural and efficient pipeline would be: **(i)** use system identification to learn approximate governing laws from scarce labeled data (a relatively fast process),  **(ii)** generate abundant synthetic data via the identified system, **(iii)** train a neural operator on this augmented dataset (very fast process). This **"system-identification-then-neural-operator”** approach is conceptually simpler, computationally cheaper, and well aligned with existing literature. Unless the authors can demonstrate that PPI-NO achieves substantial accuracy or efficiency gains over such a pipeline, the proposed coupling strategy appears unnecessarily expensive and poorly motivated.
-  **(2) Problem specificity:** The proposed framework is narrowly applicable. It only learns mappings from source terms $f$ to solutions $u$ for PDEs with at most second-order derivatives, since the surrogate network $\phi$ is designed to take inputs of $u$ and its first- and second-order derivatives. This excludes broader classes of problems, such as mappings from initial conditions or coefficients to solutions, or PDEs involving higher-order derivatives.
- **(3) Lack of interpretability or diagnostics for the learned pseudo-physics**: The paper recognizes that the learned physics network is a flexible black-box and tradeoffs interpretability for accuracy, but does not provide in-depth analysis or visual diagnostics of what types of surrogate physics are actually learned, nor how they relate to the true underlying physics when available. This could limit the practical adoption of the framework in some scientific domains.
- **(4) Unfair comparisons:** The reported advantages of PPI-NO are not convincing due to unfair experimental design. As shown in Eq. (5), PPI-NO effectively leverages additional training data from the surrogate $\phi$ ($N’$ low-fidelity pairs) beyond the $N$ high-fidelity pairs used by baselines such as FNO and DeepONet. This extra supervision is a key reason for PPI-NO’s superior performance. For fairness, either: (i) all methods should be trained with the same total amount of data (e.g., $N$ high-fidelity pairs), or (ii) baselines should be provided with analogous low-fidelity data (e.g., $N'$ low-fidelity pairs via a system-identification step). Without such adjustments, the current comparisons overstate the advantages of the proposed method.
- **(5) Idealized data assumptions:** The experiments assume access to high-fidelity ($128\times 128$ regular grids), noise-free data, which contradicts the paper’s motivating scenario. In realistic settings, data are more likely to be low-fidelity (e.g., coarse grids, sparse samples) and noisy. The framework’s robustness to these challenges is not tested.
- **(6) Missing out-of-distribution evaluation:** A key promise of operator learning is generalization across problem settings. However, the paper only tests in-distribution performance. There are no experiments on out-of-distribution generalization (e.g., extrapolating to unseen input ranges). Without this, the claims about efficiency and robustness remain incomplete.
- **(7) Missing scalability and computational cost:** Although memory costs are briefly discussed, the increased training complexity and wall-clock time due to alternating training of two networks is not analyzed quantitatively; such reporting would enhance transparency for practical use.
- **(8) Incomplete discussion of limitations**: The paper does not adequately discuss the potential drawbacks of introducing a black-box pseudo-physics network, such as risks of overfitting, spurious inductive biases, or reduced trustworthiness in scientific discovery. A more balanced discussion would help position the work more realistically.

**Audience:**

Yes

**Audience Explanation:**

The topic of enhancing neural operator performance under limited labeled data and unknown physics is timely and aligns with the interests of the TMLR community, especially those working at the intersection of machine learning and scientific computing. The proposed PPI-NO framework introduces an unconventional idea of coupling a neural operator with a learned pseudo-physics surrogate, which could stimulate discussion. However, the current limitations—restricted applicability to second-order PDEs, reliance on idealized high-fidelity data, and lack of fair baselines—significantly weaken its practical impact. While some readers may find the methodology and experimental setup interesting, the broader audience may question its novelty, fairness, and real-world relevance in its present form.

**Broader Impact Concerns:**

I do not see major ethical or societal risks tied to this work.

**Claims And Evidence:**

No

**Claims Explanation:**

The paper reports improved performance of the proposed PPI-NO framework over baseline methods, but the comparisons are not fully fair:
- (1) PPI-NO leverages additional training data (via the pseudo-physics surrogate) that the baselines do not, making it difficult to isolate the benefit of the framework itself.
- (2) Moreover, the motivation that PPI-NO is more effective than simpler alternatives, such as a “system-identification-then-neural-operator” pipeline, is not substantiated through experiments.
- (3) The method’s applicability is also restricted to PDEs with at most second-order derivatives, yet this limitation is not clearly acknowledged.
- (4) Finally, the evaluations are carried out only in idealized high-fidelity, noise-free settings, with no evidence provided for robustness to noisy data, low-fidelity observations, or out-of-distribution scenarios.

Together, these gaps reduce the strength and clarity of the evidence supporting the paper’s central claims.

**Requested Changes:**

**Critical Changes (required for acceptance)**:
- (1) **Clarify and justify the core motivation**. The authors should provide a stronger justification for why coupling a neural operator with a pseudo-physics network is necessary and beneficial, particularly in comparison to a simpler “system-identification-then-neural-operator” pipeline. A direct and fair comparison in both accuracy and efficiency would be essential.
- (2) **Fair baselines.** Ensure all comparisons are fair by using the same amount of training data for all methods. For example, if the proposed framework leverages additional pseudo-labeled data from the learned physics model, then the same number of data points should also be provided to FNO, DeepONet, or other baselines.
- (3)	**Scope of applicability.** The current method is limited to PDEs with maximum second-order derivatives and only applies to mapping source terms $f$  to $u$. The paper should clearly state this limitation and, ideally, explore extensions or discuss how the framework could handle broader classes of PDEs (e.g., systems with coefficients, higher-order derivatives, or initial-value problems).
- (4) **More realistic experiments.** Include experiments where training data is (i) available only on coarse or randomly sparse grids, and (ii) noisy or corrupted, as these reflect the very scenarios of scarce and imperfect data that motivate the paper.
- (5) **Out-of-distribution generalization.** Provide experiments to assess whether the proposed approach generalizes to inputs outside the training distribution, which is critical for operator learning tasks.
- (6) **Scalability and cost analysis.** Quantitatively analyze the runtime and memory overhead introduced by alternating optimization between two networks. This information is important for understanding the method’s practicality.

**Suggested Changes (to strengthen the work)**:
- (1) **Interpretability of pseudo-physics.** Provide illustrative examples or visual diagnostics of what the pseudo-physics network actually learns. This would help readers understand whether it captures meaningful surrogate physics or simply acts as a flexible regularizer.
- (2) **Sensitivity analysis.** Investigate the robustness of the method to variations in the architecture and parameterization of the pseudo-physics network (e.g., depth, width, inclusion/exclusion of derivative terms).
- (3) **Discussion of limitations.** Add a clear section outlining when the proposed method is most suitable and when it may not be appropriate (e.g., black-box surrogate overfitting, reduced interpretability).
- (4) **Clarify data usage.** On page 3, clarify the statement regarding decomposing a $128 \times 128$ input into 16,384 points. This does not increase information content, and a clearer explanation is needed.

---

> ### Author Response · Authors · 2025-10-18
>
> We thank the reviewer for the insightful and constructive comments. C: comments; R: response
>
> >C1: The central idea is to improve neural operator training by introducing a surrogate “pseudo-physics” network that acts as a learned PDE. However, I find the motivation weak. There already exist advanced system identification methods capable of accurately recovering governing equations from limited data. A more natural and efficient pipeline would be: (i) use system identification to learn approximate governing laws from scarce labeled data (a relatively fast process), (ii) generate abundant synthetic data via the identified system, (iii) train a neural operator on this augmented dataset (very fast process). This "system-identification-then-neural-operator” approach is conceptually simpler, computationally cheaper, and well aligned with existing literature. Unless the authors can demonstrate that PPI-NO achieves substantial accuracy or efficiency gains over such a pipeline, the proposed coupling strategy appears unnecessarily expensive and poorly motivated.
>
> R1: We agree that the suggested approach is a natural alternative. However, **system identification** methods targeting interpretability typically require **stronger assumptions and more prior knowledge** than our black-box PDE learner. For instance, the widely used **SINDy** framework [1] assumes a linear structure and depends on a predefined operator dictionary—whose choice is crucial and can easily cause model misspecification.
>
> In contrast, our **black-box PDE representation** is more flexible and assumption-light: we only specify a set of basic derivatives, while their nonlinear combinations (e.g., $\sin(u)$, $uu_x$, $u_{xx}^2$, $u_xu_y$) are learned by a neural network. This design inevitably sacrifices interpretability—there is no free lunch. Furthermore, our  **joint training** of the neural operator and the PDE representation $\phi$,allows both components to **mutually enhance** each other’s performance.
>
> To evaluate the *“system-identification–then–neural-operator”* alternative, we ran **SINDy** to first identify an approximate governing equation, used it to generate paired synthetic data, and then trained the **FNO** on the combined real and synthetic samples. The operator dictionary matched the derivative set used in our method. For comparison, we also ran our black-box $\phi$-network to generate 200 synthetic examples. The resulting test errors of the neural operator under varying training data sizes are summarized below.
>
>  *Darcy flow*:
> |  Relative L2| Training size=5  |  10 | 20  |  30 |
> |---|---|---|---|---|
> | SINDy $\rightarrow$ FNO | 0.4624(0.0147) | 0.3382(0.0113) | 0.2725(0.0115) | 0.2222(0.0033)|
> | $\phi$-network $\rightarrow$ FNO | 0.4492(0.0196) | 0.2638(0.0132) | 0.1646(0.0168) | 0.1013(0.0023)|
> |FNO| 0.4915(0.0210)  | 0.3870(0.0118)  | 0.2783(0.0212) | 0.1645(0.0071)|
> |  Ours    | 0.1716(0.0048)    | 0.0956(0.0084)   | 0.0680(0.0031)    | 0.0642(0.0010)  |
>
> *Nonlinear Diffusion*:
> |  Relative L2| Training size=5  |  10 | 20  |  30 |
> |---|---|---|---|---|
> |  SINDy $\rightarrow$ FNO| 0.2703(0.0048) | 0.2400(0.0059) | 0.1908(0.0123) | 0.1213(0.0034)|
> | $\phi$-network $\rightarrow$ FNO| 0.1603(0.0109) | 0.0943(0.0037) | 0.0540(0.0043) | 0.0361(0.0013)|
> |FNO | 0.2004(0.0083)  | 0.1242(0.0046)  | 0.0876(0.0061) |0.0551(0.0021)|
> |  Ours    | 0.0105( 0.0016) | 0.0066(0.00023) | 0.0049(0.00037) | 0.0038(0.00039) |
>
> We observe that applying **SINDy** before training **FNO** improves prediction accuracy only in the Darcy flow task with 5, 10 and 20 training examples; in all other cases, performance actually deteriorates compared to training FNO alone. This likely occurs because SINDy cannot precisely recover the true governing equations—its learned forms may be interpretable yet imperfect, producing synthetic samples that degrade FNO performance.
>
> In contrast, the **$\phi$-network → FNO** pipeline—where our $\phi$-network first generates synthetic data, followed by FNO training—consistently improves results. Although the $\phi$-network is a black-box model, it can more accurately approximate the underlying PDE structure, leading to more useful synthetic samples. Nevertheless, this sequential approach still underperforms our **jointly trained $\phi$ + FNO** model, demonstrating that **mutual optimization** between the two components further enhances FNO’s predictive accuracy.
>
>
> [1] Brunton S L, Proctor J L, Kutz J N. Discovering governing equations from data by sparse identification of nonlinear dynamical systems[J]. Proceedings of the national academy of sciences, 2016, 113(15): 3932-3937.

---

> ### Author Response · Authors · 2025-10-18
>
> >C2: The proposed framework is narrowly applicable. It only learns mappings from source terms f to solutions u for PDEs with at most second-order derivatives, since the surrogate network $\phi$ is designed to take inputs of u and its first- and second-order derivatives. This excludes broader classes of problems, such as mappings from initial conditions or coefficients to solutions, or PDEs involving higher-order derivatives.
>
> R2: Thank you for the insightful comments. We would like to clarify that our framework is **not fundamentally limited to second-order derivatives**. The use of up to second-order terms in our benchmarks was motivated by the nature of those PDEs and validated by ablation results (Table 7), which showed this was sufficient. However, the surrogate $\phi$-network is **highly flexible**—for higher-order PDEs, the input set can be easily extended to include third- or higher-order derivatives without modifying the core algorithm.
>
> Moreover, our method can also handle **mappings from coefficients to solutions**. In this setting, the $\phi$-network learns a PDE of the form
>
> $$
> L(x, u(x), S_1(u)(x), \ldots, S_Q(u)(x), f(x), S_1(f)(x), \ldots, S_Q(f)(x)) = 0,
> $$
>
> where $S_i$ are derivative operators (e.g., $u_t + \nabla(f \cdot u) = u_{xx}$). The training loss can be constructed similarly, using not only the locations in the training data but also randomly sampled input locations to ensure nontrivial $\phi$ predictions.
>
> Finally, for **mapping initial conditions to solutions**, our method remains applicable when trajectory data are available, since the PDE can be represented as
>
> $$
> \frac{\partial u}{\partial t} = g(u, t, x, S_1(u), \ldots, S_Q(u)).
> $$
>
> >C3: The paper recognizes that the learned physics network is a flexible black-box and tradeoffs interpretability for accuracy, but does not provide in-depth analysis or visual diagnostics of what types of surrogate physics are actually learned, nor how they relate to the true underlying physics when available. This could limit the practical adoption of the framework in some scientific domains.
>
>
> R3:  We thank the reviewer for raising the important point of understanding the learned surrogate physics. We agree that for scientific adoption, trust in the model is crucial. While our $\phi$-network is indeed a flexible function approximator, our primary scientific claim is not that its internal weights are interpretable, but that its overall input-output behavior accurately and robustly mimics the true physics. **In response to this comment, we add a comprehensive set of behavioral diagnostics**. This includes comparative visualizations of predicted dynamic responses under challenging initial conditions, and quantitative analyses of how the model's outputs (e.g., forces, fluxes) align with physical principles (like conservation laws) even when its internal mechanics remain a black box. This shift from "interpreting the mechanism" to "validating the behavior" provides a practical and rigorous path to building trust in scientific applications.
> To make this concrete, we ran diagnostics on two datasets: Advection and Nonlinear Diffusion. For each dataset, we evaluated the learned $\phi$-network on (i) in-distribution (ID) test fields sampled from squared-exponential (SE) Gaussian kernels with length scale 0.25, and (ii) out-of-distribution (OOD) test fields sampled from SE kernels with length scale 0.15. We also provide heatmaps to visualize the learned surrogate’s behavior.
>
> Advection heatmaps:
> https://www.dropbox.com/scl/fi/a77q6suaec3u2dlrsa2qm/advection_data_phi_network_heatmap.png?rlkey=c9wjvud23jzmsp0yedtoec1lt&st=dn51nz3h&dl=0
> Advection $\phi$-network error table:
> |  Relative L2| Training size=20  |  30 | 50  |  80 |
> |---|---|---|---|---|
> |$\phi$-network test in different distribution| 0.0092(0.0006) | 0.0071(0.0002) | 0.0060(0.0003) | 0.0047(0.0005)|
> |$\phi$-network test in same distribution| 0.0047(0.0004) | 0.0034(0.0004) | 0.0028(0.0002) | 0.0023(0.0003)|
>
> Nonlinear Diffusion heatmaps:
> https://www.dropbox.com/scl/fi/7jjf1wrsj9g3a8g28j4bt/nonlinear_diffusion_data_phi_network_heatmap.png?rlkey=pu1w4hblypzrfq42pxu05eqq5&st=hlox5g7r&dl=0
> Nonlinear Diffusion $\phi$-network error table:
> |  Relative L2| Training size=5  |  10 | 20  |  30 |
> |---|---|---|---|---|
> |$\phi$-network test in different distribution| 0.0556(0.0077) | 0.0485(0.0072) | 0.0461(0.0057) | 0.0432(0.0059)|
> |$\phi$-network test in same distribution| 0.0467(0.0065) | 0.0423(0.0049) | 0.0397(0.0033) | 0.0365(0.0043)|
>
> Across training sizes (Advection: N in {20, 30, 50, 80}; Nonlinear Diffusion: N in {5, 10, 20, 30}), the phi-network’s predictions remain closely aligned with ground truth under both ID and OOD splits, indicating robust generalization despite the black-box parameterization.

---

> ### Author Response · Authors · 2025-10-18
>
> >C4: The reported advantages of PPI-NO are not convincing due to unfair experimental design. As shown in Eq. (5), PPI-NO effectively leverages additional training data from the surrogate $\phi$ (N' low-fidelity pairs) beyond the N high-fidelity pairs used by baselines such as FNO and DeepONet. This extra supervision is a key reason for PPI-NO’s superior performance. For fairness, either: (i) all methods should be trained with the same total amount of data (e.g., N high-fidelity pairs), or (ii) baselines should be provided with analogous low-fidelity data (e.g., N' low-fidelity pairs via a system-identification step). Without such adjustments, the current comparisons overstate the advantages of the proposed method.
>
> R4: Thanks for your comments. We would like to clarify that in Eq. (5), the PDE network $\phi$ does **not** generate additional training data. Instead, $\phi$ serves to **regularize** the prediction of $\psi$—the neural operator (FNO or DeepONet). The first term in Eq. (5) represents the standard neural-operator loss for $\psi$ on the original training samples. In the second term, a new input function $f'_n$ is sampled, and its corresponding neural-operator prediction $u'_n = \psi(f'_n)$ is passed to the PDE network $\phi$ to evaluate its consistency with the PDE structure. Specifically, the $L_2$ loss $L_2(f'_n, \phi(u'_n))$ encourages the neural-operator prediction $u'_n$, together with its input $f'_n$, to conform to the PDE constraints embedded in $\phi$.
>
> In other words, $\phi$ **never explicitly generates input–output pairs** for training the neural operator.
>
> We also conducted additional experiments comparing this framework with an alternative pipeline—**system identification → synthetic data generation → neural operator training with mixed data**. Please refer to Response **R1** for the corresponding results and discussion.
>
> >C5:The experiments assume access to high-fidelity (128 x 128 regular grids), noise-free data, which contradicts the paper’s motivating scenario. In realistic settings, data are more likely to be low-fidelity (e.g., coarse grids, sparse samples) and noisy. The framework’s robustness to these challenges is not tested.
>
> R5: We appreciate the reviewer’s feedback regarding data fidelity. Our work specifically targets the challenge of **limited data availability**—that is, a small number of solution snapshots ($N$)—rather than low spatial resolution. The use of high-resolution (128×128), noise-free data in our benchmarks was intentional: it provides a controlled and stringent testbed to evaluate our method’s ability to learn a **generalizable physical model from few examples**. If the data were coarse or noisy, it would be difficult to disentangle the effect of data quality from the method’s intrinsic performance.
>
> Moreover, **high-fidelity data** is typically expensive and thus limited in quantity, whereas **low-fidelity data** is easy to acquire and rarely scarce. Hence, our setting—with few high-fidelity samples—accurately reflects the intended use case of the proposed framework.
>
> Following your suggestion, we evaluated the **robustness** of our framework to data **sparsity and noise** using the nonlinear diffusion benchmark. Each training input–output pair, originally sampled on a $128 \times 128$ noise-free grid, was randomly subsampled to 400 function values, to which we added 5% Gaussian noise. These noisy samples were then **denoised and interpolated** back to the $128 \times 128$ grid using Gaussian process interpolation (since FNO requires gridded data). We subsequently trained our **PPI-NO** model under this setting.
>
> As shown in the results below, our method **consistently outperforms FNO** across all training sizes, even in the presence of noise, though with the expected performance degradation compared to the clean-data experiments.
>
> *Nonlinear Diffusion with noise level 5% and 400 function value samples in each training pair*.
> |  Relative L2| Training size=5  | 10 | 20  | 30 |
> |---|---|---|---|---|
> |FNO| 0.2037(0.0114) | 0.1217(0.0056) | 0.0891(0.0060) | 0.0606(0.0020)|
> |Ours|0.1465(0.0062) | 0.0848(0.0041) | 0.0593(0.0023) | 0.0504(0.0026)|

---

> ### Author Response · Authors · 2025-10-18
>
> >C6:A key promise of operator learning is generalization across problem settings. However, the paper only tests in-distribution performance. There are no experiments on out-of-distribution generalization.
>
> R6: Excellent suggestion—we agree. To assess **out-of-distribution (OOD)** performance, we conducted additional experiments on the **nonlinear diffusion** and **advection** benchmarks, using test sets drawn from distributions different from the training data. Specifically, for nonlinear diffusion, we sampled input functions from a Gaussian process with a square exponential (SE) kernel of length scale is 0.1(0.2 in training data), and for advection, we generated inputs still using square exponential (SE) Gaussian kernels but length scales is 0.15(0.25 in training data). We then evaluated both our method and **FNO** on these OOD test sets.
>
> As shown below, for nonlinear diffusion, both FNO and our method achieve performance comparable to their in-distribution results, whereas for advection, performance degrades significantly for both. Nevertheless, in all cases, our method—through joint training with  the additional $\phi$-network—**consistently outperforms FNO by a substantial margin**.
>
> *OOD test: Nonleaner_diffusion*:
> |  Relative L2| Training size=5  |  10 | 20  |  30 |
> |---|---|---|---|---|
> |FNO| 0.1669(0.0084) | 0.1009(0.0012) | 0.0837(0.0016) | 0.0626(0.0009)|
> |Ours| 0.0229(0.0088) | 0.0179(0.0013) | 0.0135(0.0010) | 0.0103(0.0003)|
>
> *OOD test: Advection*:
> |  Relative L2| Training size=20  |  30 | 50  |  80 |
> |---|---|---|---|---|
> |FNO| 0.7339(0.0176) | 0.7086(0.0076) | 0.6838(0.0258) | 0.6484(0.0027)|
> |Ours| 0.6175(0.0354) | 0.6099(0.0227) | 0.6231(0.0206) | 0.6471(0.0148)|
>
> >C7: Although memory costs are briefly discussed, the increased training complexity and wall-clock time due to alternating training of two networks is not analyzed quantitatively; such reporting would enhance transparency for practical use.
>
> R7: We appreciate the reviewer's comment on training cost analysis. The alternating training between the two networks is a key design that enables the co-evolution of the solution operator and the physics surrogate. While this iterative process inevitably leads to an increase in total wall-clock time (approximately linear in the number of alternation steps), we find this cost is justified by the significant performance gains, as evidenced in our results.
>
> Here, we supplemented the wall-clock running time of our method on Darcy flow and Poisson benchmarks. The computing platform is a Linux workstation with NVIDIA GeForce RTX 4090 GPU (24G memory).
>
> standard FNO:
> |  running time(s) | Training size=5  |  10 | 20  |  30 |
> |---|---|---|---|---|
> | darcy | 14 | 14 | 14 | 15 |
> |  poisson   |  13  |  13  |  14  |  14 |
>
> PPI-FNO：
> |  running time(s) | Training size=5  |  10 | 20  |  30 |
> |---|---|---|---|---|
> | darcy | 984 | 1001 | 1036 | 1064 |
> |  poisson   |  869    | 881   |  908   |  935  |
>
> >C8: The paper does not adequately discuss the potential drawbacks of introducing a black-box pseudo-physics network, such as risks of overfitting, spurious inductive biases, or reduced trustworthiness in scientific discovery. A more balanced discussion would help position the work more realistically.
>
> R8: We appreciate the reviewer's suggestion for a more balanced perspective. We agree. Here is our discussion of the limitations of our current work.
>
> First, the inherent risk of the $\phi$-network overfitting to limited data and producing unphysical models, despite our regularization efforts.
>
> Second, the challenge of verifying that the learned physics is genuine and not spurious correlations, underscoring the need for rigorous out-of-distribution testing.
>
> Third, the implications of using a black-box model for scientific discovery, where interpretability is traded for predictive performance, and how this affects the model's role as a tool for hypothesis generation rather than law discovery.
> This discussion will help readers better understand the appropriate contexts for applying our framework.

---

> ### Author Response · Authors · 2025-10-22
>
> >C1: Clarify and justify the core motivation.  The authors should provide a stronger justification for why coupling a neural operator with a pseudo-physics network is necessary and beneficial, particularly in comparison to a simpler “system-identification-then-neural-operator” pipeline. A direct and fair comparison in both accuracy and efficiency would be essential.
>
> Thank you for your suggestion, we have added related experiment and analysis in **section 5.2, starting from the middle of page 12**. We also discussed it before in **R1** of the rebuttal.
>
> >C2:  Fair baselines. Ensure all comparisons are fair by using the same amount of training data for all methods. For example, if the proposed framework leverages additional pseudo-labeled data from the learned physics model, then the same number of data points should also be provided to FNO, DeepONet, or other baselines.
>
> Thank you for the comments. We discussed this in **rebuttal R4**, and we have added the clarification in **section 3.2, page 6 , around line 30**.
>
> >C3: Scope of applicability. The current method is limited to PDEs with maximum second-order derivatives and only applies to mapping source terms f to u. The paper should clearly state this limitation and, ideally, explore extensions or discuss how the framework could handle broader classes of PDEs (e.g., systems with coefficients, higher-order derivatives, or initial-value problems).
>
> We thank the reviewer for raising the important point of understanding the learned surrogate physics. We have talked about it in rebuttal **R2**, we have added the explanation with blue color in **section 5.2, page 16 in "Ablation study on the choice of derivatives."**
>
> >C4: More realistic experiments. Include experiments where training data is (i) available only on coarse or randomly sparse grids, and (ii) noisy or corrupted, as these reflect the very scenarios of scarce and imperfect data that motivate the paper.
>
> Thanks for your comments. We have added the related experiment in **section 5.2, page 18, "ablation study on the noise data."** and responded in rebuttal **R5**.
>
> >C5:Out-of-distribution generalization. Provide experiments to assess whether the proposed approach generalizes to inputs outside the training distribution, which is critical for operator learning tasks.
>
> We appreciate the reviewer’s feedback, We have made changes in **section 5.2, page 16 "Ablation study on the out-of-distribution."** and discussed the topic in rebuttal **R6**.
>
>
> >C6:Scalability and cost analysis. Quantitatively analyze the runtime and memory overhead introduced by alternating optimization between two networks. This information is important for understanding the method’s practicality.
>
> Excellent suggestion, answered in rebuttal **R7**, and we also made changes and added tables in **section 5.2, page 20, line 1 with blue color in "Memory and Time Cost."**
>
> >C7:  Interpretability of pseudo-physics. Provide illustrative examples or visual diagnostics of what the pseudo-physics network actually learns. This would help readers understand whether it captures meaningful surrogate physics or simply acts as a flexible regularizer.
>
> We appreciate the reviewer's comment. We have responded in rebuttal **R3**. We also added the heat map and tables to show the result in **section 5.2, page 16 "Ablation study on the out-of-distribution."**
>
> >C8: Sensitivity analysis. Investigate the robustness of the method to variations in the architecture and parameterization of the pseudo-physics network (e.g., depth, width, inclusion/exclusion of derivative terms).
>
> Thanks for the opinion. We have actully done some related experiments about derivative terms in table 8, which describes the influence of inclusion/exclusion of derivative orders. Moreover, we also set experiment and get results shown in Table 7, which states the influence of the architecture of $\phi$.
>
> >C9: Discussion of limitations. Add a clear section outlining when the proposed method is most suitable and when it may not be appropriate (e.g., black-box surrogate overfitting, reduced interpretability).
>
> We appreciate the reviewer's suggestion. We responded in rebuttal **R8**. We have made changes in **section 7 on page 21** and talk about limitations.
>
> >C10:  Clarify data usage. On page 3, clarify the statement regarding decomposing a 128x128 input into 16,384 points. This does not increase information content, and a clearer explanation is needed.
>
> Thank you for your suggestion. I have added a brief explanation on **page 3 with blue color before equation 3**.

---

### Review · Reviewer_7jtP · 2025-10-04

**Summary Of Contributions:**

In this paper, the authors introduce a Pseudo Physics-Informed Neural Operator (PPI-NO) framework --- the goal of operator learning is to approximate an operator that maps input parameters and/or functions to corresponding output functions.
- For one example, such scenarios may arise in the context of solving partial differential equations.
- Specifically, this paper aims to learn the solution operator of a PDE system, using a training dataset, which includes different instances of the PDE system sampled at a set of locations.

The concept of physics-informed neural operator (PINO) aims to embed physical laws, i.e., the governing equations, into the learning process.

This paper is motivated by the case when physics knowledge is absent, hence it is impossible to construct the physics loss term as in (PINO).

As a result, this paper proposes to design a neural network to approximate the general form of the PDE system. An $\ell_2$ is used to reconstruct the parameters of the neural network. Next, the paper also proposes to leverage the pseudo-physics laws to enhance neural operator learning.

Finally, this paper reports experiments that evaluate their proposed method on five commonly used benchmark operator learning problems in the literature, as well as an application of fatigue modeling.

**Audience:**

Yes

**Audience Explanation:**

The paper concerns learning neural networks to approximate the outcome of a PDE system. This may be related to applying ML to science broadly speaking but the connection is superficial and I think more examples and applications are required to further substantiate the contribution to the machine learning audience.

**Claims And Evidence:**

No

**Claims Explanation:**

My understanding of this paper is that it proposes to use a neural network to approximate the outcomes of a PDE system, the mechanism for doing this is based on a literature called physics-informed neural operators. The main idea here is to approximate physical laws by closely tracking their outcomes in a surrogate model.

- However, I find the connection to "physical laws" to be unconvincing --- I did not see either theoretical results or empirical evidences showing that the proposed modeling is capable of following a "physical law."

- The proposed algorithm is more like a Monte-Carlo approximation algorithm to approximate the instances from the input function space given enough samples. However, the paper does not state the sample size requirement for achieving this accurately. In the worst-case scenario, imagine learning a PDE whose derivative is specified by a piecewise linear function. The sample complexity would scale with the number of pieces, to say the least, as a lower bound. [Ref: Nearly-tight VC-dimension and Pseudodimension Bounds for Piecewise Linear Neural Networks. JMLR 2019]

Therefore, the proposed algorithm in its full generality seemed hopeless but the paper does not specifiy the conditions or assumptions under which the algorithm is expected to work well.

- The experiments including the detailed ablation studies are interesting. However, in table 2, it wasn't clear to me how to calibrate the reported numbers (are the reported results good enough for the proposed problem? many are in the range of 60% to 70%). As a result I was unable to evaluate the strengths of the empirical results. [Disclaimer: I have limited experience working in the area of "physics-informed neural operators" so my judgment may be biased here]

- Finally, I think the paper could benefit from some additional editing and polishing as I noted plenty of typos and misformated content. For instance, the figures in Figure 8 and 9 should be downsized. Table 8 is placed right after the end of the conclusion.

The proposed algorithm could use a better explanation.

**Requested Changes:**

There is plenty of room for improvement in this paper. Here are several examples (but there are probably more than the items below that need to be addressed):

- First, the paper requires a major revision to improve its formatting and its explanation of the main concepts.

- Second, the paper needs to explain the conditions or assumptions under which their proposed algorithm will work more explicitly.

- Third, I think the paper can add more examples and applications to substantiate why the proposed method is relevant and interesting to the machine learning audience.

- Finally, it would be helpful to add a paragraph summarizing the contribution to the literature in the introduction.

---

> ### Author Response · Authors · 2025-10-18
>
> We thank the reviewer for the constructive comments. C: comments; R: response
>
> >C1: I find the connection to "physical laws" to be unconvincing --- I did not see either theoretical results or empirical evidences showing that the proposed modeling is capable of following a "physical law."
>
> R1: The meaning of *physical laws*, in our view, is twofold:
>
> (1) they possess an **equation-based representation**, typically PDEs, and
>
> (2) they carry an **interpretable physical meaning**.
>
> We never claim our work discovers true physical laws. Our $\phi$-network merely provides a **neural PDE representation** without interpretability, hence the term *“pseudo physics.”*
>
> Our goal is not to recover real equations but to introduce a **PDE-structured inductive bias** that improves generalization—consistent with physics-informed learning literature, where models are trained to satisfy given PDE structures for enhanced predictive accuracy.
>
> **Empirical Evidence.**
> Tables 1–2 show consistent and substantial error reduction when incorporating our black-box PDE representation ($\phi$-network). When the neural operator learns to follow the “pseudo physics” imposed by $\phi$, prediction errors decrease significantly.
>
> **Theoretical Evidence.**
> The $\phi$-network, as a neural network, has sufficient representation power to approximate any PDE operator of the form
> $$L(x, u(x), S_1(u)(x), \ldots, S_Q(u)(x)),$$
>
> where $S_i$ denotes derivatives, e.g., $L = u_{xx} + u u_x + u^2 - 1 + \cos(u)$.
> Because real PDEs are smooth analytic functions of (u) and its derivatives, the **universal approximation theorem** guarantees that such mappings can be accurately approximated by neural networks:
> $$f(x) = \sum_{i=1}^N a_i , \sigma(w_i^\top x + b_i),$$
> for any continuous $f: \mathbb{R}^n \to \mathbb{R}$, given sufficient neurons and a non-polynomial activation (e.g., ReLU, sigmoid, tanh)(Cybenko1989,Hornik1991).
>
> [Cybenko1989] Cybenko G. Approximation by superpositions of a sigmoidal function[J]. Mathematics of control, signals and systems, 1989, 2(4): 303-314.
>
> [Hornik1991] Hornik K. Approximation capabilities of multilayer feedforward networks[J]. Neural networks, 1991, 4(2): 251-257.
>
> >C2: The proposed algorithm is more like a Monte-Carlo approximation algorithm to approximate the instances from the input function space given enough samples. However, the paper does not state the sample size requirement for achieving this accurately. In the worst-case scenario, imagine learning a PDE whose derivative is specified by a piecewise linear function. The sample complexity would scale with the number of pieces, to say the least, as a lower bound.
>
> R2: Our method employs **centered finite differences** to approximate the derivatives of $u$, a classical scheme with well-established accuracy and sample complexity. The approximation error scales as $O(h^2)$ for all derivative orders, where $h$ is the grid spacing. To achieve an error $\le \epsilon$, the required number of grid points is $n = \Theta(\epsilon^{-d/2})$, where $d$ is the input dimension.
>
> For the neural network that approximates the PDE form, the **sample complexity** $m(\epsilon, \delta)$ is the smallest number of samples $m$ ensuring, with probability at least $1-\delta$,
> $$
> |R(h) - \hat{R}(h)| \le \epsilon, \quad \forall h \in \mathcal{H},
> $$
>
> where $R(h)$ and $\hat{R}(h)$ denote generalization and training errors, respectively.
> From computational learning theory,
> $$
> m(\epsilon, \delta) = O\left(\frac{d_{VC} + \log(1/\delta)}{\epsilon^2}\right),
> $$
> where $d_{VC}$ is the VC dimension of the network. For a network with $W$ parameters and piecewise-linear activations (e.g., ReLU), $d_{VC} = O(W \log W)$; for smooth activations, $d_{VC} = O(W^2)$.
>
> >C3:The experiments including the detailed ablation studies are interesting. However, in table 2, it wasn't clear to me how to calibrate the reported numbers (are the reported results good enough for the proposed problem? many are in the range of 60% to 70%). As a result I was unable to evaluate the strengths of the empirical results. [Disclaimer: I have limited experience working in the area of "physics-informed neural operators" so my judgment may be biased here]
>
> R3: Thank you for the careful reading. To clarify, the values in Table 2 report relative error reduction (%) against the baseline, not absolute accuracy. Thus, **entries in the 60–70% range mean our framework reduces the baseline error by 60–70% (i.e., substantially improves accuracy)**. For example, if a baseline error were 0.10, a 60% reduction corresponds to an error of 0.04; a 70% reduction corresponds to 0.03. Higher percentages are better.

---

> ### Author Response · Authors · 2025-10-22
>
> >C1: First, the paper requires a major revision to improve its formatting and its explanation of the main concepts.
>
> Thank you for your suggestion. As we answered in **rebuttal R1 and R2**, we have also added more explanation and experiment in updated manuscript **section 3, starting from page 4, and section 5.2 from page 12 with blue color.**
>
> >C2: Second, the paper needs to explain the conditions or assumptions under which their proposed algorithm will work more explicitly.
>
> We have added more ablation study and limitation discuss. Please see **section 5.2 on middle of page 12 and section 7 on page 21**.
>
> >C3: Third, I think the paper can add more examples and applications to substantiate why the proposed method is relevant and interesting to the machine learning audience.
>
> We admit that we don't have enough time prepare more different realistic PDE data experiments. However, we added more experiments about the ablation study to strengthen our robustness in applications further in **section 5.2 with blue-colored paragraphs.**
>
> >C4: Finally, it would be helpful to add a paragraph summarizing the contribution to the literature in the introduction.
>
> We have already summarized our contribution at the end of the introduction in our original manuscript on page 2.

---

### Decision · Action_Editor_ygHs · 2025-10-31

**Recommendation:** Accept with minor revision

**Additional Comments:**

The submission is well motivated and thoroughly validated. Remaining points that should be addressed for a camera-ready version:

- Reviewer `oRPB` suggests acceptance "provided that the new findings and the authors’ responses in the comment section are incorporated into the manuscript."
- Reviewer `SQyJ` raises three remaining concerns that should be addressed:
    * In the extension for learning from coefficients to solutions, explain how the proposed training loss avoids or mitigates the non-convexity and identifiability issues arising from multiple coefficient-solution pairs yielding zero loss.
    * Provide a clear justification for why using a larger input distribution in the PDE network does not constitute an unfair advantage over baselines, or otherwise adjust comparisons accordingly.
    * Explicitly explain how the PDE network functions as a regularizer without effectively introducing additional data or supervision beyond what baselines use.
- In general, align figure styles: make axis labels larger (especially in Fig. 10), legend text smaller (especially in Fig. 9), and ensure that all figures are legible and of high quality.

**Audience:**

Yes

**Audience Explanation:**

The work on machine learning for scientific discovery and operator learning offers a creative bridge between physics-informed and data-driven paradigms. Readers in scientific ML, PDE learning, etc., will find results valuable, even if theoretical validation is still pending.

**Claims And Evidence:**

Yes

**Claims Explanation:**

The paper presents extensive experiments and ablation studies. The results consistently demonstrate the claims. While one reviewer is of a different opinion, the majority of reviewers agree that the claims are supported by evidence.

---

> ### Author Response · Authors · 2025-11-13
>
> We thank the editor for the decision and constructive comments. C: comments; R: response
>
> # for editor ygHs
>
> >C1: Reviewer oRPB suggests acceptance "provided that the new findings and the authors' responses in the comment section are incorporated into the manuscript."
>
> R1: Thank you for your comments. As we answered in rebuttal, we have incorporated all our responses, experiment results, and analysis into our manuscript. We also point out the paper changes locations as a list here:
>
> - findings about out of distribution: section 5.2, page 17 "Ablation study on the out-of-distribution"
>
> - discussion about incorporating parameters, sources, and IC/BC: section 6.1, page 21.
>
> - discussion about data independence: section 3.1(page 5) and in section 6.2(page 22).
>
> - findings about system-identification–then–neural-operator:  in section 5.2, page 13. "Step-by-step approach: system-identification-then-neural-operator."
>
> - clarification about fair comparisons: in section 3.2, bottom of page 6.
>
> - findings about noisy data: in section 5.2, bottom of page 17, "ablation study on the noise data."
>
> - findings experiment about time cost: in section 5.2, page 21.
>
> - clarification of data usage: page 3 before equation 3.
>
> - discussion about limitations: in section 6.2 (page 22).
>
> >C2: In the extension for learning from coefficients to solutions, explain how the proposed training loss avoids or mitigates the non-convexity and identifiability issues arising from multiple coefficient-solution pairs yielding zero loss.
>
> R2: We thank the reviewer for pointing out the potential non-convexity and identifiability issues when learning mappings from coefficients to solutions. We agree that, if the PDE residual is enforced only on a fixed, finite set of collocation points, there can be multiple coefficient–solution pairs that achieve (near-)zero loss, and the most *trivial* degenerate solution is a PDE network that predicts zero residual everywhere. To address this issue, we can use two approaches.
>
> First, in our setting, we design the PDE loss so that the residual is small on correctly paired coefficient–solution samples and *large* on randomly *mismatched* pairs drawn from the same coefficient distribution. Concretely, in the coefficient-to-solution setting we assume that true training pairs (u,f) approximately satisfy a PDE of the form
> $$
> L(x, u(x), S_1(u)(x), \ldots, S_Q(u)(x), f(x), S_1(f)(x), \ldots, S_Q(f)(x)) = 0,
> $$
>
> For such correctly paired data (u,f), we encourage a small residual, where the $L^2(\Omega)$ norm is approximated by averaging over randomly sampled locations x in the domain $\Omega$. In contrast, if we keep the same solution $u$ but pair it with an independent random coefficient $\tilde f$ drawn from the same coefficient distribution, we expect that
> $$
> L^2(x, u(x), S_1(u)(x), \ldots, S_Q(u)(x),\tilde f(x), S_1(\tilde f)(x), \ldots, S_Q(\tilde f)(x)) > 0,
> $$
>
> This suggests a contrastive objective:
> $$
> \mathcal{J}(\theta) = L^2(x, u(x), S_1(u)(x), \ldots, S_Q(u)(x), f(x), S_1(f)(x), \ldots, S_Q(f)(x)) - L^2(x, u(x), S_1(u)(x), \ldots, S_Q(u)(x),\tilde f(x), S_1(\tilde f)(x), \ldots, S_Q(\tilde f)(x)),
> $$
>
> where each $L^2(\Omega)$ norm is approximated by averaging the squared residual over randomly sampled locations in the same domain $\Omega$ as the data. The first term enforces a small residual on true coefficient–solution pairs, while the second term explicitly penalizes zero (or very small) residuals on randomly mismatched pairs. Under this objective, a constant-zero PDE network is no longer optimal: the model can strictly decrease $\mathcal{J}(\theta)$ by keeping the residual small on true pairs but making it large on random mismatches, which reduces degeneracy and improves identifiability in practice.
>
> Second, we can also use likelihood-based generative models such as PixelCNN[1], learning to maximize the probability density of coefficients and solutions at the sampled locations. The probability density is parameterized by the network $\phi$.  This is complementary to the residual-based objective above.
>
> We have added the related discussion in section 6.1 (page 21).
>
> [1] Aron van den Oord, Nal Kalchbrenner, Lasse Espeholt, Oriol Vinyals, Alex Graves, and Koray Kavukcuoglu. Conditional image generation with pixelcnn decoders. In Advances in Neural Information Processing Systems, 2016.

---

> ### Author Response · Authors · 2025-11-13
>
> >C3: Provide a clear justification for why using a larger input distribution in the PDE network does not constitute an unfair advantage over baselines, or otherwise adjust comparisons accordingly.
>
> R3: We thank the reviewer for raising this concern. We clarify that in all experiments, the PDE network is trained on exactly the same input distribution as the main operator network and all baselines. We do not extrapolate beyond the domain, nor do we change the distribution over coefficients / initial conditions.
>
> When we mention “randomly sampled input” these are sampled within the same domain and distribution as the model inputs and no solutions, and their role is simply to *densify* the evaluation of the PDE, not to give the PDE network access to additional regions of the input space. In other words, the PDE network does not enjoy a larger input distribution; it only uses more collocation points drawn from the same support. Moreover, since we only sampled the inputs data, it cannot be used in original neural operator model as feature and label pairs.
>
> In addition, Table 7 provides an ablation where we  simply employ a FNO as a drop-in replacement of the $\phi$-network, trained under the same input distribution. This variant is consistently much worse than our design, indicating that the performance gains do not come from giving the PDE network a larger or richer input distribution, nor simply from higher model capacity, but rather from the specific PDE-structured regularization we propose.
>
> We have clarified these points in the revised manuscript in section 5.2 "Ablation study on pseudo physics network $\phi$"(page 15) to avoid any misunderstanding about the training distribution used for the PDE network and the fairness of our comparisons.
>
> >C4: Explicitly explain how the PDE network functions as a regularizer without effectively introducing additional data or supervision beyond what baselines use.
>
> R4: We thank the reviewer for this question and clarify that the PDE network acts purely as a model-based regularizer, without introducing any additional labeled data or supervision beyond what baselines use. In practice, it is realistic to sample new input $f'$ from the same input distribution, but it is not realistic to obtain corresponding ground-truth solutions $u'$; thus, we never observe extra $(f', u')$ pairs. Instead, we sampled $f'$, we then first pass it through the current operator network $\psi$ to obtain a predicted solution $\hat u'$, then feed $(\hat u', x)$ into the PDE network $\phi$ to produce a reconstructed source term $\tilde f''$. We then use the discrepancy $| \tilde f'' - f' |_2$ as a regularization term, as formalized in Eq. (5). This “PDE consistency” term depends only on (i) the same labeled training pairs $(f_i, u_i)$ that baselines use to train $\psi$ and $\phi$, and (ii) unlabeled source term $f'$ drawn from the same input distribution. It therefore does not add new supervision, but simply encourages $\psi$’s predictions to lie on the manifold of solutions that are self-consistent with the learned PDE representation, functioning as a regularizer rather than as extra data.
>
> We have added the clarification in section 3.2(page 6) after equation 5.
>
> >C5: In general, align figure styles: make axis labels larger (especially in Fig. 10), legend text smaller (especially in Fig. 9), and ensure that all figures are legible and of high quality.
>
> R5: Thank you for your suggestion. We have updated our figures to make it clearer.